# Interpretable Embeddings of Speech Explain and Enhance the Brain Encoding Performance of Audio Models [*]

## Abstract

Speech foundation models (SFMs) are increasingly hailed as powerful computational models of human speech perception. However, since their representations are inherently black-box, it remains unclear what drives their alignment with brain responses. To remedy this, we built linear encoding models from six interpretable feature families: mel-spectrogram, Gabor filter bank features, speech presence, phonetic, syntactic, and semantic features, and contextualized embeddings from three state-of-the-art SFMs (Whisper, HuBERT, WavLM), quantifying electrocorticography (ECoG) response variance shared between feature classes. Variance-partitioning analyses revealed several key insights: First, the SFMs' alignment with the brain can be mostly accounted for by their ability to learn and encode simple interpretable speech features. Second, SFMs exhibit a systematic trade-off between encoding of brain-relevant low-level and high-level features across layers. Finally, our results show that SFMs learn brain-relevant semantics which cannot be accounted for by lower-level speech features, with this capacity increasing with model size and context length. Together, our findings suggest a principled approach to build more interpretable, accurate, and efficient encoding models of the brain by augmenting SFM embeddings with interpretable features.

## 1 Introduction

Large-scale foundation models have become increasingly popular tools for studying how the brain processes language and speech. Trained on vast amounts of linguistic and acoustic data, these models appear to extract rich features relevant for predicting neural activity. Numerous studies have employed large language models (LLMs) and demonstrated their effectiveness in predicting neural responses to narrative stimuli by constructing linear models based on their internal representations (Toneva & Wehbe, 2019; Goldstein et al., 2022; Caucheteux & King, 2022; Antonello et al., 2023; Hosseini et al., 2024; Mischler et al., 2024). Similarly, speech foundation models (SFMs) have been used to investigate neural representations underlying speech perception. These studies commonly conclude that, like LLMs, SFMs are highly effective in predicting brain responses to speech due to their ability to integrate contextual information over time (Vaidya et al., 2022; Millet et al., 2022; Anderson et al., 2024; Goldstein et al., 2025; Tuckute et al., 2023; Antonello et al., 2023).

Despite these promising results, the neuroscientific basis of SFM's predictive performance remains to be thoroughly investigated. Much of the existing literature has focused on predictive success, offering limited insight into which specific representational components drive their alignment with the brain. Although recent work (Anderson et al., 2024; Oota et al., 2024) has begun to address this question, the findings have been inconclusive: for example, while Oota et al. (2024) argue that SFM lacks brain-relevant semantic information, Anderson et al. (2024) provide evidence suggesting the presence of brain-relevant lexical encoding. In parallel, complementary work in machine learning has examined SFM's internal representations using probing and representational similarity analysis (Pasad et al., 2022; 2023; 2024; Ashihara et al., 2023; Shen et al., 2023; Choi & Yeo, 2022; Choi et al., 2024). Results from this line of work indicate that SFM representations encode a wide range of features: early layers often reflect acoustic and phonetic properties, whereas

---

[*]Disclosure: Portions of this manuscript were assisted by OpenAI ChatGPT for writing. The authors reviewed, edited, and take full responsibility for the content.

later layers increasingly gain high-level syntactic and semantic information (Pasad et al., 2022; 2023; 2024). While these studies have revealed important aspects of SFM representations, they do not test which specific representational components actually drive SFMs' brain encoding performance, and thus leave a critical gap in our understanding of SFM-brain alignment.

Due to this gap, several key questions regarding the nature of SFM-brain alignment remain unanswered. For example, several studies report an almost monotonic increase in brain encoding performance across SFM layers, with the latest layers in the network substantially outperforming earlier layers (Vaidya et al., 2022; Antonello et al., 2023; Anderson et al., 2024). A key open question is why this trend occurs. Does it reflect a genuine accumulation of increasingly high-level, brain-relevant information, with low-level details retained to support neural encoding? Or does it mask a trade-off, where gains in high-level representations come at the expense of lower-level features? Even more fundamentally, although the main representational advantage of SFMs is that they are not constrained to an interpretable feature space, it remains unclear how much benefit this provides beyond simpler interpretable features. The central question, then, is whether SFMs' brain encoding performance primarily reflects their capacity to capture speech information in highly non-linear and uninterpretable ways, or instead derives largely from their ability to learn and encode simpler, interpretable features.

This study addresses these two central questions of SFM-brain alignment: How much of an SFM's predictive power is accounted for by interpretable features? Across layers, is there a trade-off between encoding of low- and high-level brain-relevant information? To answer these, we utilized intracranial electrocorticography (ECoG), which has high temporal and spatial resolution that makes it suitable for studying both low-level and high-level feature encoding. We analyze this data with a variance partitioning approach applied to hand-crafted, interpretable features spanning acoustic to lexical domains. Our analysis reveals: First, the SFMs' alignment with the brain can be mostly accounted for by their ability to learn and encode simple, interpretable speech features. Second, over layers, SFMs exhibit a loss in brain-relevant low-level acoustic information with the corresponding increase in high-level information. Finally, our results show compellingly that SFMs learn brain-relevant semantics that cannot be accounted for by lower-level speech features, with this capacity increasing with model size and context length. Together, these findings suggest a principled approach to building more interpretable, accurate, and efficient encoding models of the brain by augmenting SFM embeddings with interpretable features.

## 2 Methods

Fig. 1 presents a summary of the methodology employed in this research.

### 2.1 Dataset

All results were derived using open-source data contained in the Podcast ECoG Dataset (Zada et al., 2025). It contains electrocorticography (ECoG) recordings from nine patients with epilepsy who listened to a 30-minute excerpt of the podcast *This American Life* ("So a Monkey and a Horse Walk Into a Bar: Act One—Monkey in the Middle"). Of the 1,330 intracranial electrodes originally implanted, 1,268 were retained after excluding contacts with imprecise anatomical localization or excessive noise in their power-spectral density. Electrode positions were projected onto a template cortical surface for visualization. Full pre-processing details are provided in Zada et al. (2025).

We focused on broadband high-gamma activity (70–200 Hz), extracted with a fourth-order zero-phase Butterworth band-pass Infinite Impulse Response (IIR) filter and converted to analytic amplitude. The resulting envelopes were down-sampled to 4 Hz and z-scored.

### 2.2 Interpretable Features

We derived six families of interpretable predictors: (i) mel-spectrograms, (ii) Gabor filter bank (GBFB) features, (iii) speech presence, (iv) phonetic features, (v) syntactic features, and (vi) semantic features (Fig. 1a). These features reflect the language processing hierarchy in speech comprehension (Heer et al.,

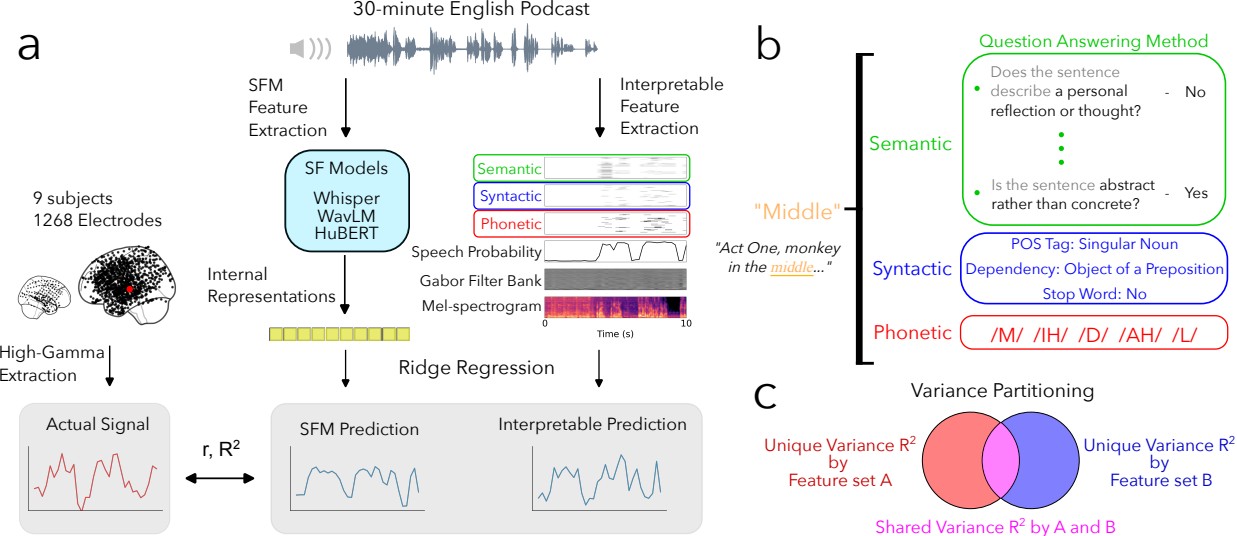

Figure 1: *Summary of the methodology.* (a) We constructed six sets of interpretable features—mel-spectrograms, Gabor filter bank features, speech presence, phonetic, syntactic, and semantic features—and compared them to representations from three state-of-the-art speech foundation models (SFMs): Whisper, HuBERT, and WavLM. The comparison is based on each representation's ability to predict neural responses, measured through Pearson's $r$ and the variance accounted for $R^2$. (b) An example of each linguistic feature set. The word is represented as phonetic (ARPAbet), syntactic (Part-of-Speech Tag, Dependency Parsing, stop word or not), and semantic (Question Answering) features. (c) Diagram of the variance partitioning approach.

2017; Keshishian et al., 2023). As shown in Fig. S1, each feature accounts for unique neural variance $R^2$ relative to the others, demonstrating that they capture complementary aspects of brain responses to speech.

**(i) Mel-spectrograms.** We computed mel-spectrograms from 0–8 kHz with 32 mel bins and further converted them to decibels using `librosa` (McFee et al., 2015). To create a feature vector for time $t$ s, we concatenated the frames spanning $[t - 0.5, t]$s, yielding a 672-dimensional vector. Dimensionality was reduced by principal-component analysis (PCA), retaining 95% of the variance.

**(ii) Gabor filter bank (GBFB) features.** Spectro-temporal modulation features were extracted with the Gabor filter-bank implementation of Schädler et al. (2012). The 455-dimensional features, computed at 100 Hz, were down-sampled to 4 Hz using `scipy.signal.resample` (Virtanen et al., 2020) to align with the neural data; PCA (95% variance) was applied prior to modeling.

**(iii) Speech presence.** Speech probability was estimated with EfficientAT, a CNN-based audio tagger (Schmid et al., 2023). To produce a feature scalar for time $t$ s, we fed the model the audio spanning $[t - 0.5, t]$s.

**(iv) Phonetic features.** Each word in the transcript was converted to its ARPAbet phoneme representations using `nltk.corpus.cmudict` (Bird & Loper, 2004). Word intervals were evenly subdivided among their $N$ phones, assigning on- and offset times to every phoneme. Every phoneme was one-hot encoded over the 39 ARPAbet symbols. To create a feature vector for time $t$ s, the embeddings for every phoneme whose offset time lies in $[t - 0.5, t]$s were summed, producing a $T \times 39$ matrix.

**(v) Syntactic features.** Following Zada et al. (2025), each word was first converted into 96-dimensional embeddings, comprising 50 part-of-speech tags, 45 dependency labels, and a stop/non-stop label using the `spaCy` (Honnibal et al., 2020). To create a feature vector for time $t$ s, the embeddings for every word whose offset time lies in $[t - 0.5, t]$s were summed, producing a $T \times 96$ matrix. Fig. S2 shows the histogram of number of words in 0.5-second windows. Notably, the rare instances of 5 words in a 0.5-second window (0.3%) are attributable to moments of overlapping speech between multiple speakers.

**(vi) Semantic features.** Following Benara et al. (2024), we obtained 168-dimensional question-answering (QA) semantic embeddings by prompting GPT-4 (`gpt-4-0125-preview`; OpenAI (2023)) with 56 yes/no questions about each word's meaning (e.g., *"Does the word/sentence involve an expression of personal values or beliefs?"*) with 3 different context lengths (word-level, 1.5 s, and 3 s). For the 1.5/3-second context length, the model was additionally provided with the words spoken in the 1.5/3 seconds preceding each target word when answering the questions. To create a feature vector for time $t$ s, the embeddings for every word whose offset time lies in $[t - 0.5, t]$s were summed, producing a $T \times 168$ matrix. Because of how this feature is constructed, its effective context window can extend up to approximately 4.75 seconds—accounting for the 3-second input context, the additional 0.5-second summing window, and the maximum word duration (about 1.25 seconds for words like "approximately"). To ensure reproducibility, we provide the full list of questions in Table S2 and include the generated answers in the supplementary material.

## 2.3 Speech Models

For this work, unless otherwise specified, we used the largest variants of three speech models: Whisper Large V1 (638M Parameters, 32 layers) (Radford et al., 2022), HuBERT X-large (964M Parameters, 48 layers) (Hsu et al., 2021), and WavLM Large (317M Parameters, 24 layers) (Chen et al., 2022). This choice was motivated by prior findings from Antonello et al. (2023), which showed that the brain encoding performance of SFM models increased logarithmically with the number of model parameters. Table S1 shows the summary of the architecture and training objectives for each model. To generate the embeddings with a context length of $Ctx$ seconds, we applied a sliding window of $Ctx$ seconds and a stride of 250 ms to the input audio. For each window, the representation at a given layer was defined as the hidden state of the final token. We made this choice to remain consistent with prior work (Vaidya et al., 2022; Antonello et al., 2023; Oota et al., 2024) and to maximize interpretability, since the encoding results depend on a single causal token from each layer, making them easier to compare and interpret. For Whisper, we only used hidden states from the encoder. In this study, $Ctx$ was varied across the following values: 0.5, 1, 2.5, 5, 10, 20, and 30 seconds. PCA with 95% variance was applied to each model's embeddings to reduce dimensionality.

## 2.4 Encoding Model Construction

After processing, the high gamma signals we predict have shape $[T, N_{ch}]$ where $T$ is the number of time points (i.e., 7200) and $N_{ch}$ is the number of channels (i.e., 1268). To estimate the model performance, we used a 5-fold cross-validation scheme, splitting the signals into 5 temporally contiguous chunks. For each fold, the model was trained on 80% of the data and tested on the remaining 20%. Within the training folds, we employed a 4-fold nested cross-validation scheme to optimize the model parameters. For features requiring dimensionality reduction, PCA parameters were estimated from the training set and applied to the test set to ensure no data leakage occurred.

Prior to conducting variance partitioning, we computed encoding performance values from embeddings with different context lengths for each model (Fig. S3). The context lengths yielding the highest test correlations across layers were 30 seconds for HuBERT, 5 seconds for WavLM, and 30 seconds for Whisper. Unless otherwise noted, all subsequent analyses are based on embeddings with these optimal context lengths.

### 2.4.1 Banded ridge regression

To create a joint encoding model with features from different feature spaces and dimensionalities, we employed banded ridge regression, which has been shown to improve neural encoding performance when using diverse feature spaces (Nunez-Elizalde et al., 2019). Unlike ridge regression with spherical Gaussian priors, banded ridge regression optimizes the regularization parameter separately for each feature space. We used the `himalaya` library by Dupré la Tour et al. (2022). The regression formulation becomes:

$$\boldsymbol{\beta}^{\star} = \arg \min_{\boldsymbol{\beta}} \left\| \sum_{i=1}^{m} \mathbf{X}_i \boldsymbol{\beta}_i - \mathbf{y} \right\|_2^2 + \sum_{i=1}^{m} \alpha_i \left\| \boldsymbol{\beta}_i \right\|_2^2 .$$

Here, $\mathbf{X}_i \in \mathbb{R}^{T \times p_i}$ is the feature *matrix* for the $i$-th feature space, $\boldsymbol{\beta}_i \in \mathbb{R}^{p_i}$ is the corresponding *vector* of regression weights, $\mathbf{y} \in \mathbb{R}^T$ is the observed response *vector* (e.g., neural responses), $\alpha_i \in \mathbb{R}_{\geq 0}$ is the *scalar* regularization parameter for the $i$-th feature space, and $m$ is the (scalar) number of feature spaces. The optimal coefficient vector concatenated across spaces is $\boldsymbol{\beta}^\star = \left[ \boldsymbol{\beta}_1^\top, \ldots, \boldsymbol{\beta}_m^\top \right]^\top \in \mathbb{R}^p$ with $p = \sum_{i=1}^m p_i$.

Each feature was z-scored before applying the regression model. Unless otherwise specified, we varied alpha over 20 logarithmically spaced points between $10^0$ and $10^3$. When using a single feature space ($m = 1$), banded ridge regression simplifies to standard ridge regression, as the optimization reduces to finding a single scalar regularization parameter.

### 2.4.2 Performance Metrics

To measure the performance of the encoding models, we used correlation (Pearson's $r$) and the variance accounted for $R^2$, two standard metrics for evaluating encoding models (Vaidya et al., 2022; Anderson et al., 2024; Goldstein et al., 2025; Tuckute et al., 2023; Li et al., 2023). Generally, we used Pearson's $r$ to compare the encoding model performance, while the variance accounted for $R^2$ was used to quantify the unique and shared contributions of different features for neural prediction. Negative $R^2$ values were clipped to 0, since negative variance lacks interpretability, following Tuckute et al. (2023); Hadidi et al. (2025); Anderson et al. (2024).

### 2.4.3 Variance Partitioning

To quantify the shared and unique variance contributions of feature sets, we used the variance partitioning method established by Heer et al. (2017). Between two feature sets $\mathcal{A}$ and $\mathcal{B}$,

$$R^2_{\text{unique}(\mathcal{A})} = R^2_{\mathcal{A},\mathcal{B}} - R^2_{\mathcal{B}} \qquad R^2_{\text{shared}(\mathcal{A},\mathcal{B})} = R^2_{\mathcal{A}} + R^2_{\mathcal{B}} - R^2_{\mathcal{A},\mathcal{B}}$$

where $R^2_{\mathcal{A}}$, $R^2_{\mathcal{B}}$, and $R^2_{\mathcal{A},\mathcal{B}}$ denote the variance accounted for by models using only $\mathcal{A}$, only $\mathcal{B}$, and both $\mathcal{A}$ and $\mathcal{B}$ together, respectively.

### 2.4.4 Computational Resources

All experiments and analyses were conducted on AMD EPYC 7662 CPUs and NVIDIA L40 GPUs, except for the generation of QA semantic features, which was performed using the OpenAI API. Cumulatively, the experiments required approximately 500 GPU hours.

## 3 Results

### 3.1 Decomposing SFM Brain Encoding Performance with Interpretable Features

Fig. 2 decomposes SFM encoding model performance into variance accounted for by interpretable features versus unaccounted for by them, and examines this decomposition from three perspectives (Fig. 2a): (i) To what extent can the neural variance accounted for by SFMs be attributed to interpretable features? (ii) How does the shared neural variance between SFMs and individual feature sets change across model layers? (iii) If there is any neural variance that cannot be attributed to known interpretable features, how does it evolve across model layers? Does it scale with the number of model parameters?

**Most of the neural variance captured by SFMs is attributable to interpretable features.** Fig. 2b shows the ratio of neural variance shared between the SFM representations and a combined set of all interpretable features, relative to the total variance accounted for by the SFM representations. This metric directly quantifies what proportion of the variance captured by SFMs can be attributed to known, interpretable features. This ratio is particularly high in early layers, exceeding 95% in the earliest layers of these models. Even among the latest layers, these proportions remained substantially high at above 75% respectively. These results suggest that in early layers, nearly all the neural variance captured by SFMs can be accounted for by interpretable features. Although the ratio decreases in later layers, it stays substantial, implying that interpretable confounds continue to account for much of the neural variance.

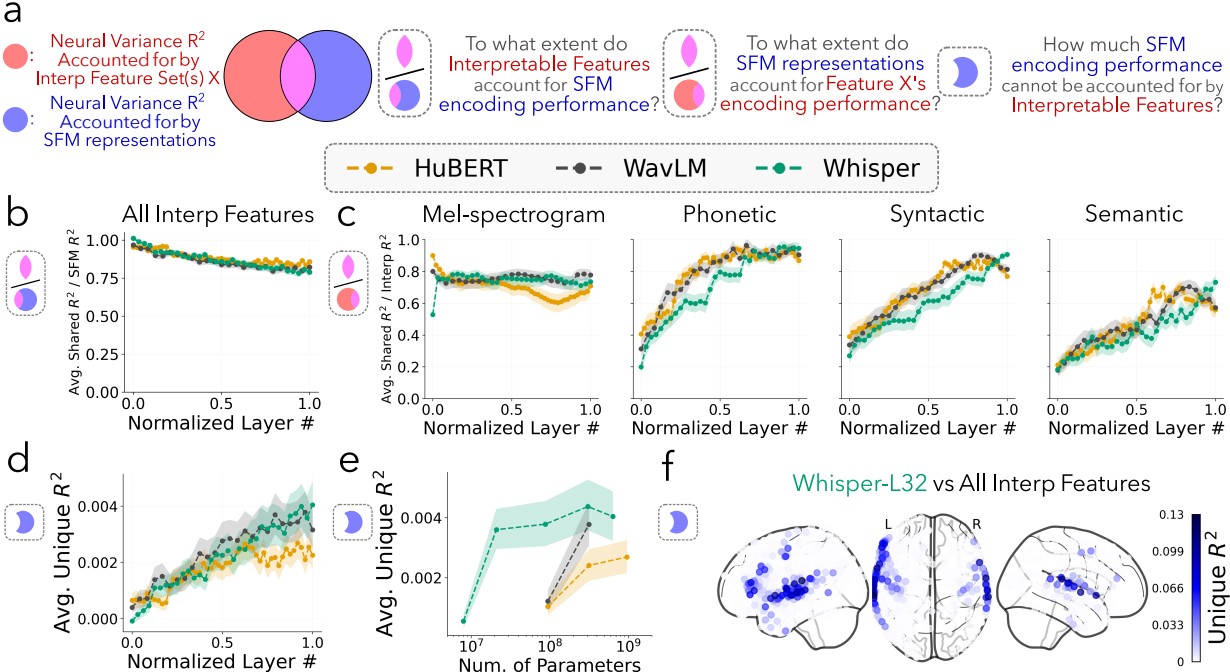

Figure 2: *Decomposing SFM brain encoding performance with interpretable features.* (a) Diagram of the variance partitioning analysis used to answer three key questions about the encoding model performance of SFMs. (b) The ratio of neural variance shared by the SFM representations and the combined set of all interpretable features, relative to the total variance accounted for by the SFM representations. In early layers, the interpretable model accounts for nearly all of the neural variance, and even in later layers, it continues to account for a substantial majority. (c) The ratio of neural variance shared between different interpretable feature sets and the SFM representations, relative to the total variance accounted for by each interpretable feature set. This ratio peaks in early layers for the mel-spectrogram, but in progressively later layers for higher-level features (phonetic, syntactic, and semantic). (d) Unique neural variance of SFMs with respect to the combined set of all interpretable features, averaged across all electrodes. For all three models, the increase is monotonic over layers. (e) Unique neural variance of SFMs as a function of model size. For each model size, the layer with the highest unique variance was selected. Results show that the SFMs' unique neural variance increases with model size. (f) Unique neural variance accounted for by the embeddings from Whisper layer 32 with respect to the combined set of all interpretable features, plotted on a template brain. The largest unique contributions appear in the superior temporal gyrus (STG) and inferior frontal gyrus (IFG), regions that are typically associated with the processing of higher-order information. In panels (b–e), shaded regions indicate 95% confidence intervals estimated via bootstrapping.

**SFMs display a layerwise tradeoff between encoding of brain-relevant low-level and high-level features.** Fig. 2c displays the ratio of neural variance shared between each interpretable feature set and the SFM representations, relative to the total variance accounted for by that feature set alone. This ratio indicates how well the speech model captures the neurally relevant information contained in that feature space. Here, we report results for the mel-spectrogram, phonetic, syntactic, and semantic features; results for additional feature sets are provided in Fig. S5.

The ratios exhibited distinct trajectories for low-level versus high-level features. For the mel-spectrogram, the ratio peaked in early layers—at layer 0 for HuBERT (0.90) and WavLM (0.80), and at layer 5 for Whisper (0.79). In later layers, the ratios decreased, reaching the lowest values of 0.61, 0.73, and 0.71, respectively. While the decrease was modest for WavLM and Whisper, further analysis in Section 3.3 suggests that this reflects a trade-off: although their embeddings lose precision, they concurrently gain broader temporal windows of acoustic information. A similar decreasing trend was observed for other low-level features, such

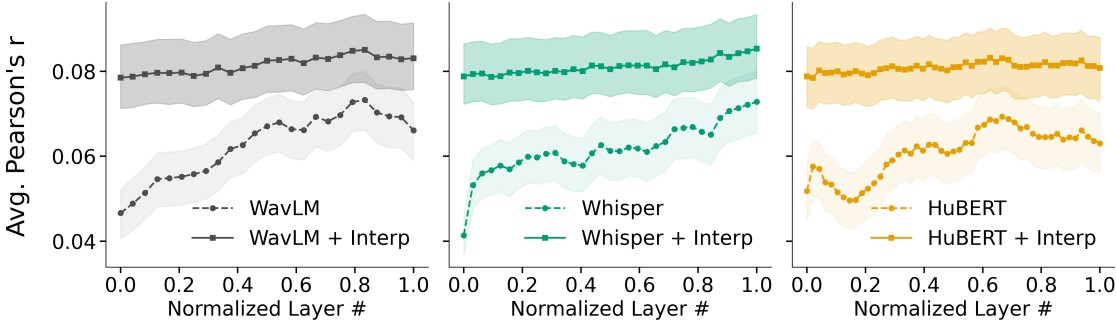

Figure 3: *Interpretable features efficiently enhance the encoding model performance of SFMs.* Test correlation value for each SFM and its joint model with the combined set of all interpretable features, averaged across electrodes. Shaded regions denote 95% confidence intervals estimated via bootstrapping. *Left*: WavLM, *Middle*: Whisper, *Right*: HuBERT. Combining interpretable features with SFMs offers two key advantages: (1) it compensates for the fact that no single SFM layer captures all information relevant for neural processing, and (2) it leverages the increasing amount of unaccounted variance in deeper SFM layers, which also scales with model size. Results show that incorporating interpretable features consistently improves encoding performance across all layers (p < 0.001, Wilcoxon signed-rank test).

as GBFB features and speech probability (see Fig. S5). In contrast, alignment with higher-order features emerged progressively across layers. For phonetic features, all three models showed a sharp increase after the initial layers, reaching near-maximal ratios (0.95 for HuBERT, 0.96 for WavLM, and 0.95 for Whisper) in middle-to-late layers, where they remained consistently high thereafter. Syntactic and semantic features exhibited a more gradual rise, with the ratios increasing steadily across layer depth and peaking in later layers than phonetic features. Together, these results indicate a trade-off in the neurally relevant components of SFM representations: *gains in high-level representations emerge at the expense of lower-level features, rather than through a simple monotonic accumulation of information.*

**Unique neural variance of SFMs grows with layer depth and scale.** Fig. 2d shows the unique neural variance accounted for by the SFMs relative to the combined set of all interpretable features, averaged across electrodes. This unique variance increased monotonically across layers in all three models. For HuBERT, it peaked in the late-middle layers and remained high thereafter, whereas for WavLM and Whisper, the peak occurred in the final layers. This shows that the later layers of SFMs are the most useful for capturing components of neural variance that cannot be accounted for by interpretable counterparts. Fig. 2e shows the average unique neural variance of SFMs as a function of model size. Unique variance increased systematically with the number of model parameters, indicating that the it scales with model size. This result aligns with Antonello et al. (2023), which found that the total neural variance accounted for by SFMs increases logarithmically with the number of parameters.

Fig. 2f shows the unique neural variance accounted for by the final layer of Whisper, relative to the combined set of all interpretable features, projected onto a template brain. The largest unique contributions appear in the superior temporal gyrus (STG) and inferior frontal gyrus (IFG), regions that are typically associated with processing of higher-order information (Newman et al., 2003; Vigneau et al., 2006; Friederici et al., 2000; Keshishian et al., 2023). Similar spatial patterns are observed for HuBERT and WavLM, as presented in Fig. S6. Combined with earlier findings that most neural variance in the early layers can be accounted for by interpretable features, these results suggest that the remaining unique variance in later layers likely reflects high-level, non-linear representations that are not captured by the current interpretable feature sets.

## 3.2 Building More Interpretable and Accurate Brain Encoding Models by Augmenting SFM representations with Interpretable Features

Our findings in the previous section reveal two key insights. First, there is a trade-off between low-level and high-level feature encoding in SFM representations across layers. This suggests that no single layer

Table 1: Comparison of encoding model performance (Pearson's r) across different baselines and the joint model of interpretable features and SFM embeddings. The joint model consistently outperforms all baselines.

| Model | Baseline 0 (1 Layer, 1 SFM) | Baseline 1 (Early+Late SFM Layers) | Baseline 2 (Two SFMs) | Our Method (SFM+Interp) |
|---|---|---|---|---|
| Whisper | 0.073 | 0.075 | 0.075 | **0.085** |
| HuBERT | 0.069 | 0.072 | 0.075 | **0.083** |
| WavLM | 0.073 | 0.074 | 0.074 | **0.085** |

contains all the neural-relevant features necessary for modeling speech perception. Second, deeper layers account for additional neural variance that is not captured by our interpretable features, and this unique contribution increases with model size. These observations motivate a principled approach for building more interpretable and accurate encoding models: combining interpretable feature sets with the later layers of large SFMs. This strategy mitigates the limitations of relying on individual layers while harnessing the representational advantages conferred by scaling, resulting in a more comprehensive and effective neural encoding model for speech.

Fig. 3 shows the average Pearson's $r$ for models based on SFM representations alone and for the joint model combining SFM representations with interpretable features. For all models, incorporating interpretable features consistently improves encoding performance across all layers ($p < 0.001$, Wilcoxon signed-rank test). Crucially, the source of this improvement can be traced to specific feature sets by the variance partitioning analysis in Fig. 2c. The gains are driven by the unique variance in interpretable features that SFM embeddings fail to capture: specifically, while SFMs effectively encode phonetic and syntactic information in deeper layers, they do not fully capture low-level acoustic and high-level semantic features, which the interpretable features complement.

To further validate these gains, Table 1 compares the joint model against two robust baselines: (1) a multi-layer baseline combining the 0th and best-performing layer of each SFM (Layer 32 for Whisper/HuBERT, Layer 20 for WavLM), and (2) a multi-model baseline combining the best-performing layers from pairs of SFMs (e.g., Whisper+HuBERT). The joint model consistently outperforms both baseline approaches. Importantly, the joint models also simplify interpretation. For WavLM and HuBERT, correlations based solely on SFM representations peak in mid-to-late layers but then decline toward the final layers. When interpretable features are incorporated, this drop-off is substantially reduced as the information lost in the final layers is encoded by the interpretable feature sets. In the joint model, encoding performance always reflects contributions from interpretable features spanning low-level acoustic to high-level lexical domains, while the unique neural variance accounted for by the SFM representations grows with depth. This results in a more interpretable and accurate model of the brain.

### 3.3 Trade-off Between Precision and Temporal Windows of Acoustic Information Encoded by SFMs

Our earlier finding in Section 3.1—that the neural variance shared between the mel-spectrogram and Whisper or WavLM decreases only modestly across layers—is puzzling at first sight. This stability contrasts with evidence from Pasad et al. (2022; 2023) indicating that speech foundation models progressively lose fine acoustic details as information propagates through their transformer layers. To reconcile this apparent discrepancy, we tested a new hypothesis: the last token embeddings of Whisper and WavLM encode short, precise windows of mel-spectrogram in early layers, but shift toward encoding longer windows with less precision in later layers, maintaining the overall shared variance.

To test this hypothesis, we quantify the shared neural variance between Whisper representations and mel-spectrogram features with different window lengths (Fig. 4a). Here, we present results for Whisper; results for WavLM and HuBERT are provided in Fig. S7. Fig. 4b shows the average shared neural variance $R^2$ between Whisper representations and mel-spectrograms computed using four temporal windows (0.05, 0.1, 0.3, and 0.5 seconds), across the whole brain as well as in Heschl's gyrus (HG), the Insula, and the superior temporal gyrus (STG). Importantly, the gap between each successive lines shows changes in shared variance

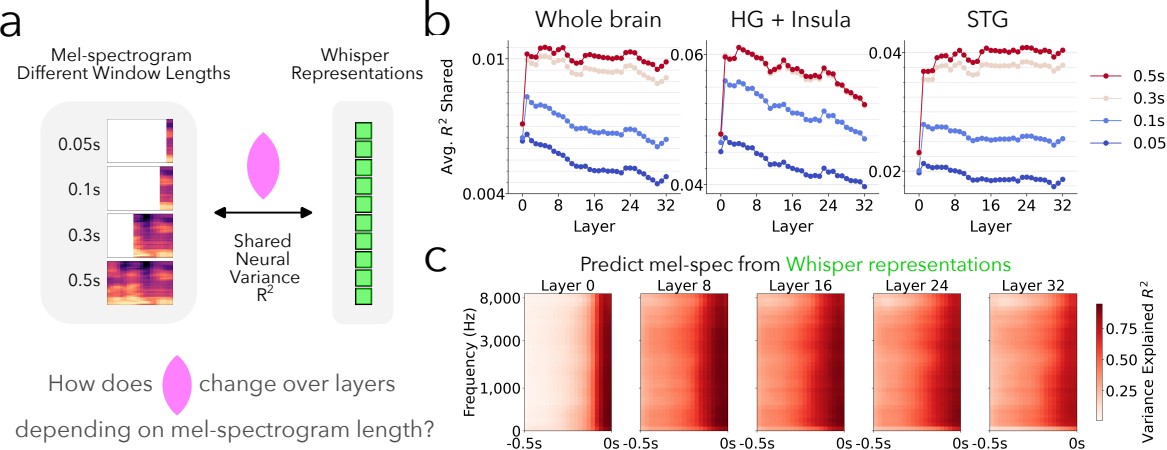

Figure 4: *Whisper loses fine-acoustic details and gains longer acoustic temporal windows in later layers.*
(a) We quantify the shared neural variance between Whisper representations and mel-spectrogram features
with different window lengths. (b) Average neural variance $R^2$ shared between Whisper representations and
mel-spectrogram segments of different temporal windows. Across all anatomical regions, shared variance
with the short window $[t - 0.05, t]$s consistently decreases across layers, whereas shared variance with the
longer window $[t - 0.5, t]$s exhibits anatomically dependent patterns. (c) Variance $R^2$ of the mel-spectrogram
$[t - 0.5, t]$s accounted for by the Whisper representations over different layers. Early layers account for a
high proportion of variance for short temporal windows; conversely, later layers account for less variance but
reflect information from broader temporal windows.

as the window length increases, quantifying how much Whisper representations capture the neural variance
uniquely attributable to mel-spectrogram information within specific temporal windows. As shown in the
plot, the shared variance decreases monotonically with the shortest window ($[0, 0.05]$s) and increases with
longer windows ($[0.1, 0.5]$s), suggesting that Whisper embeddings progressively lose fine, short-timescale
acoustic detail while gaining longer-timescale acoustic information.

Notably, different anatomical regions display distinct trends. In HG and the Insula, the shared variance
decreases across layers even when using the 0.5-second mel-spectrogram, whereas in STG, it increases over
layers until it plateaus at around layer 16. For STG, the gain in shared variance with longer windows
($[0.1, 0.5]$s) outweighs the loss observed with shorter ones ($[0, 0.05]$s), while the opposite is true for HG and
the Insula. This likely reflects differences in their temporal integration windows, which have been shown
to lengthen from primary to non-primary auditory areas (Lerner et al., 2011; Sabat et al., 2025; Norman-
Haignere et al., 2022). HG, with its short integration window, is sensitive to fine acoustic detail, while STG,
with its longer integration window, prioritizes broader temporal patterns.

To further test our hypothesis, we directly predicted 0.5-second segments of the mel-spectrogram from the
SFM representations. Specifically, we used the representations at time $t$ to predict the corresponding mel-
spectrogram within the $[t-0.5, t]$s window. Fig. 4c shows the variance $R^2$ of mel-spectrogram accounted for by
Whisper representations across different layers. In the 0th layer, Whisper representations accurately predict
all frequency bins for the shorter segments of the mel-spectrogram. However, as information progresses
through the network, the variance accounted for in these short segments decreases. At the same time, the
predictions become temporally broader, indicating that deeper layers shift toward encoding longer-range
temporal structure at the expense of fine-grained acoustic detail. This pattern provides further support for
the hypothesis that the last token embeddings of Whisper encode short, precise windows of mel-spectrogram
in early layers, but shift toward encoding longer windows with less precision in later layers.

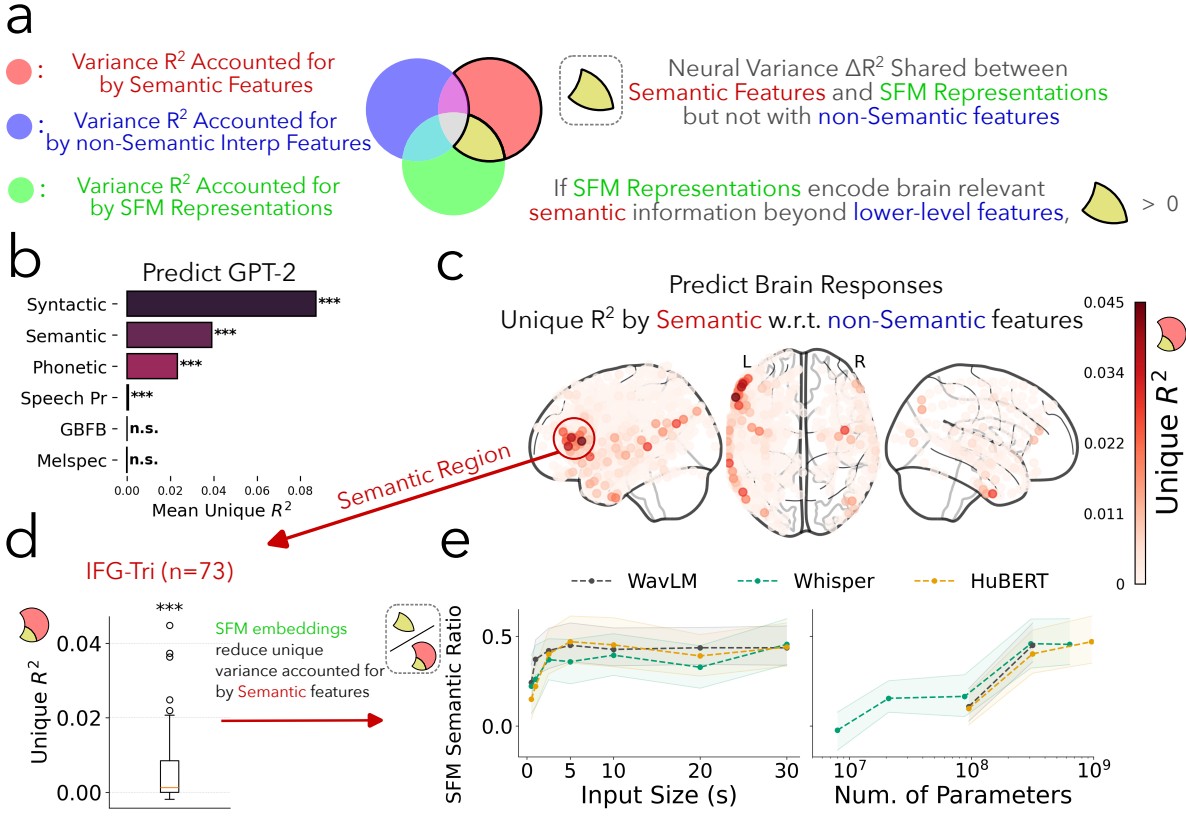

Figure 5: *SFMs encode brain-relevant semantic information.* (a) Diagram of the three-way variance partitioning analysis used to assess whether SFM representations encode semantic information. (b) Unique variance accounted for $R^2$ by different feature sets in predicting GPT-2 representations, averaged across GPT-2's feature dimensions ($p < 0.001$, Wilcoxon signed-rank test). The significant unique contribution of our question-answering semantic features confirms that they encode distinct semantic information. (c) Unique neural variance accounted for by semantic QA features relative to non-semantic interpretable features. (d) Box plot of the values in (c), across electrodes in the inferior frontal gyrus triangularis (IFG-Tri). Median values were significantly greater than zero ($p < 0.001$, Wilcoxon signed-rank test). (e) The proportion of unique semantic neural variance accounted for by SFM embeddings, averaged across electrodes in IFG-Tri. Shaded regions denote 95% confidence intervals estimated via bootstrapping. *Left:* Varying context lengths, using the largest variant of each SFM. *Right:* Varying model sizes, with context length fixed to 5 seconds for WavLM and HuBERT, and 30 seconds for Whisper. It shows SFMs encode brain-relevant semantic information that cannot be reduced to lower-level features, improving with context length and model size.

## 3.4 SFMs Encode Brain-Relevant Semantic Information

Fig. 5a outlines our approach for evaluating the degree to which SFM representations encode brain-relevant semantic information. First, we calculate the unique neural variance $R^2$ accounted for by semantic QA features with respect to non-semantic interpretable feature sets (mel-spectrogram, GBFB, speech presence, phonetic, and syntactic features). We then assess how much this unique neural variance is reduced when SFM representations are included in the model alongside the non-semantic features. This reduction reflects the shared neural variance $\Delta R^2$ between semantic QA features and SFM representations that is not accounted for by non-semantic interpretable feature sets.

Before we assess SFMs, to evaluate whether our QA features actually encode semantic information, we computed the unique variance $R^2$ by each interpretable feature set in predicting GPT-2 representations (Fig. 5b). This step was included to ensure the quality of QA features, which depend on the accuracy of

the LLM's answers. Syntactic, semantic, and phonetic features account for the largest proportion of unique variance ($p < 0.001$, Wilcoxon signed-rank test) across GPT-2's feature dimensions. Speech probability also accounts for a statistically significant amount of unique variance, though its contribution is markedly smaller in comparison. In contrast, GBFB and mel-spectrogram features do not account for any significant unique variance, as expected—GPT-2 is not trained on acoustic input and thus should not encode such information. The significant unique variance accounted for by our semantic features supports the claim that they indeed capture meaningful semantic content.

Next, to test whether semantic information captured by QA features is relevant to neural processing, we computed the variance of the brain responses uniquely accounted for by semantic features relative to non-semantic interpretable features (Fig. 5c). Semantic features had the largest unique variance accounted for in the inferior frontal gyrus pars triangularis (IFG-Tri) - a part of Broca's area that was previously found to be important for processing the semantic meaning of words (Newman et al., 2003; Vigneau et al., 2006; Friederici et al., 2000). As shown in Fig. 5(d), the median of the unique neural variance accounted for by semantics in IFG-Tri was significantly above 0 ($p < 0.001$, Wilcoxon signed-rank test). These results confirm that our semantic QA features are not only semantically meaningful but also relevant to brain regions involved in semantic comprehension.

The left plot of Fig. 5e shows the ratio of unique neural variance of semantic features shared with SFM embeddings, relative to the total unique neural variance accounted for by semantic features. Averaged across all electrodes in IFG-Tri, this ratio was significantly above 0 ($p < 0.001$, Wilcoxon signed-rank test). For WavLM and HuBERT, it increases up to a 5-second context window, consistent with the 4.75-second intrinsic context of our semantic QA features. For Whisper, the increase extended up to 30 seconds, potentially because Whisper was trained strictly with 30 seconds of audio. The right plot in Fig. 5e shows the same metric, but for varying model sizes with fixed context size. Here, the ratio increases with model size, suggesting that larger models encode more brain-relevant semantic information. Fig. S8 depicts this ratio layer-by-layer, revealing that brain-relevant semantic encoding consistently increases with depth, a trend that parallels the findings in Fig. 2b.

## 4 Discussion

In this work, we examined the encoding model performance of speech foundation models (SFMs) for human auditory and language processing. By systematically comparing the largest variants of three state-of-the-art models (Whisper, HuBERT, and WavLM) with six classes of handcrafted acoustically/linguistically interpretable features in their ability to predict ECoG responses, our analysis advances the understanding of how SFMs encode speech information relevant for neural processing. Crucially, this work addresses several previously unresolved aspects of SFM encoding performance, clarifying their representational alignment with human perceptual and linguistic processes.

First, we demonstrated that the vast majority of neural variance captured by speech foundation models (SFMs) is attributable to interpretable features—nearly all in the earliest layers, and still a substantial portion (approximately 80%) in the deeper layers. This finding suggests that much of the observed SFM–brain alignment reflects well-characterized acoustic and linguistic structure. In a parallel line of work, Hadidi et al. (2025) reported similar results in the context of LLMs, showing that most of the neural variance accounted for by GPT2-XL could be accounted for by simple confounds such as word-rate and positional information. Together, these results indicate that much of the apparent model–brain alignment across modalities can be accounted for by straightforward, interpretable features. The unique neural variance accounted for by SFMs—beyond what is accounted for by interpretable features—is primarily localized to the superior temporal gyrus and inferior frontal gyrus, and increases with model size. Moving forward, it will be important to precisely characterize the nature of this residual variance, for example by applying linear probing methods to the residual SFM representations after regressing out interpretable features.

Second, our analyses revealed a systematic trade-off in how SFMs encode neurally relevant information across layers: the shared neural variance with low-level features (mel-spectrogram, GBFB) peaked in early layers, while it increased monotonically with higher-order features (phonetic, syntactic, semantic) over layers. A closer look at mel-spectrogram encoding showed that Whisper and WavLM progressively lose fine-grained

acoustic detail but expand their temporal receptive windows with depth, whereas HuBERT exhibits wide windows from layer 0. We hypothesize that these differences stem from model architectures: HuBERT's convolutional relative positional embeddings give tokens long receptive windows from layer 0; WavLM's additional gating mechanism likely accounts for its divergence from HuBERT; and Whisper's use of fixed sinusoidal encodings directly on log-mel spectrograms constrains early receptive windows, forcing them to expand only in later layers. Future research should extend this analysis to a diverse group of model architectures to thoroughly test this hypothesis.

Third, whether SFMs encode brain-relevant semantics is debated, with evidence supporting (Pasad et al., 2022; Chen et al., 2024; Anderson et al., 2024) and challenging (Choi et al., 2024; Oota et al., 2024). Our method employs an interpretable semantic embedding to directly test for the shared neural variance with SFMs, while controlling for lower-level acoustic and linguistic information. We show that SFM representations significantly reduce the unique variance accounted for by the semantic embedding, providing strong evidence that they indeed encode brain-relevant semantic information. Importantly, our findings help to reconcile the conclusions of Oota et al. (2024), who argued that the SFMs lack brain-relevant semantics. We showed that SFMs' capacity to encode brain-relevant semantics follows a scaling law (Fig. 5) and peaks in the final layers (Fig. 2c). Oota et al. (2024) used embeddings from intermediate layers of Whisper-small, a configuration where we observe minimal semantic encoding, likely leading them to underestimate the model's capabilities at scale. Nonetheless, an important nuance remains: SFMs account for only about 40% of the unique variance accounted for by the semantic features in IFG, with respect to non-semantic features, indicating that their semantic representations remain partial. Recent studies (Moussa et al., 2025; Vattikonda et al., 2025) have shown that fine-tuning SFMs on fMRI data can improve their alignment with language regions of the brain. Thus, while our findings affirm the presence of semantic information in SFM representations, they also suggest room for improvement.

Lastly, we showed that augmenting SFM representations with interpretable features significantly improves brain encoding model performance. This strategy overcomes the limitations of relying on individual SFM layers by integrating complementary information from hand-crafted features while still benefiting from the representational advantages of model scaling. Importantly, we do not rule out the possibility that SFM representations alone could achieve comparable performance. Indeed, alternative approaches have aggregated embeddings from multiple tokens or combined representations across layers, as in prior work (Goldstein et al., 2022). However, such strategies are substantially less efficient. For example, because Whisper processes audio as log-mel spectrograms with positional encodings, representing a 0.5-s segment requires approximately 25 tokens (each covering 20 ms). For Whisper-large, this translates to $25 \times 1{,}280 = 32{,}000$ dimensions. By contrast, our interpretable feature set achieves the same encoding with fewer than 1,500 dimensions (672 for the mel-spectrogram alone), while capturing all of the neural variance accounted for by early SFM layers. We observed that even after applying PCA, the dimensionality of the Whisper embeddings remains more than an order of magnitude higher. In other words, using interpretable features provides a more compact and interpretable representation of the same information, making them a principled complement to SFM embeddings.

**Limitations.** 1. Our findings are based on data from nine clinical ECoG participants who listened to a single 30-min English podcast. While the high temporal resolution of ECoG and broad electrode coverage across brain regions make this dataset particularly well-suited for the present study, replication using larger and more diverse datasets, including other modalities such as EEG and fMRI, will be important for assessing the generalizability of these results. 2. While we controlled for several lower-level features in our analysis of semantic encoding in SFMs, we cannot entirely rule out the possibility that unaccounted confounds contribute to the shared variance between SFM representations and our semantic features. However, the observed effect of context size provides some evidence against this interpretation, as lower-level features typically operate over shorter integration windows and are therefore less sensitive to extended context. 3. While our interpretable features account for most of the neural variance captured by SFM representations, a statistically significant portion remains unaccounted for. Identifying the nature of this residual variance is an important direction for future research. 4. While there are different classes of models that can take speech as input, including AudioLLMs, we restricted our analyses to SFMs. The rationale is twofold: first, most current AudioLLMs are modular systems that pair an audio encoder—often Whisper or a self-supervised learning model—with

a language model, so their audio representations largely build on the same foundations we study. Second, AudioLLMs are primarily optimized for interactive or task-specific use cases, such as question-answering or dialogue, rather than for producing stable, general-purpose audio features, making them less directly suited for the type of neural encoding comparisons considered in this work. Nonetheless, future work may explore how AudioLLMs, given their rapid development and growing capabilities, could provide complementary insights into neural processing of speech.

**Broader Impact Statement**

This study relies on electrocorticography (ECoG) data collected from patients with epilepsy who underwent invasive monitoring for clinical purposes. While the dataset we analyze is fully de-identified and publicly available, it is important to acknowledge that these recordings originate from a vulnerable patient population. The use of such data requires careful attention to ethical considerations, including the assurance that appropriate informed consent and clinical oversight were in place during data collection, and that secondary analyses such as ours respect the dignity and autonomy of the participants.

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

# A    Appendix

## A.1    Unique Neural Variance by Each Interpretable Feature

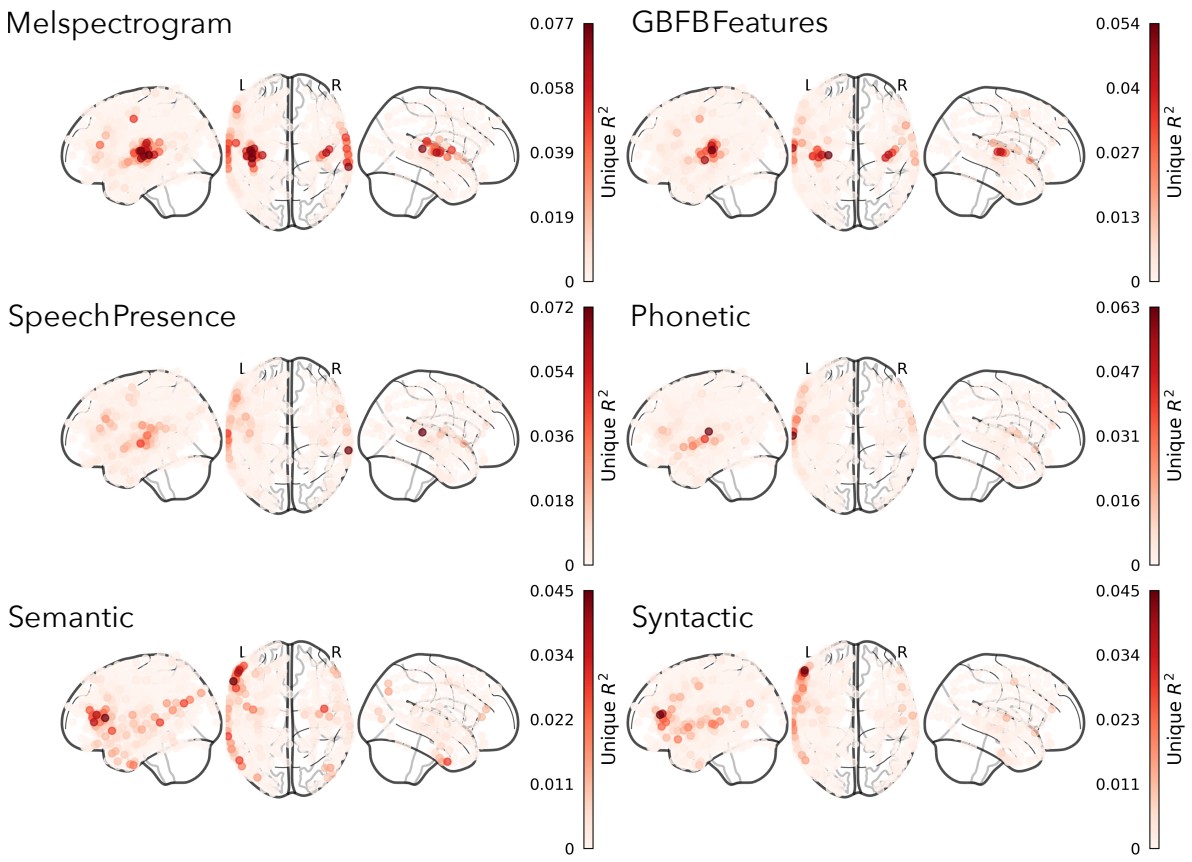

Figure S1: Unique neural variance $R^2$ accounted for by each feature with respect to the rest of the interpretable features. The language processing hierarchy can be clearly observed in the anatomical distributions.

## A.2    Word Count in a 0.5-second Window

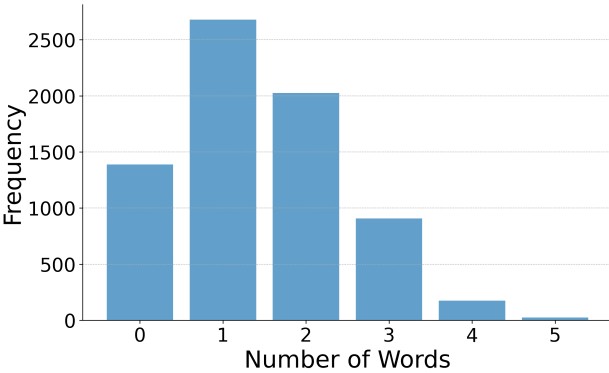

Figure S2: The histogram of number of words in a 0.5-second window used to obtain linguistic features.

### A.3 SFM Feature Extraction

Representations from speech foundation models (SFMs) were extracted using the code provided by Antonello et al. (2023) (available at `https://github.com/HuthLab/encoding-model-scaling-laws`). The input audio file, totaling 1800 seconds, was divided into five contiguous 360-second segments before being fed into the SFMs. This segmentation was performed to prevent data leakage between training and test folds in the 5-fold cross-validation scheme.

### A.4 Effect of context length on brain prediction performance

Among the context sizes tested, the context size with the highest average correlation was 30 seconds for Whisper and HuBERT, and 5 seconds for WavLM (Fig. S3). This variability highlights the need for caution when interpreting the influence of context length on encoding performance. For instance, Anderson *et al.* found that EEG encoding performance with Whisper peaked at context lengths of 5–10 seconds and interpreted this as evidence that EEG preferentially reflects the encoding of speech within that temporal window. However, it is essential to distinguish whether such patterns reflect inherent properties of neural processing or are instead artifacts of model-specific biases.

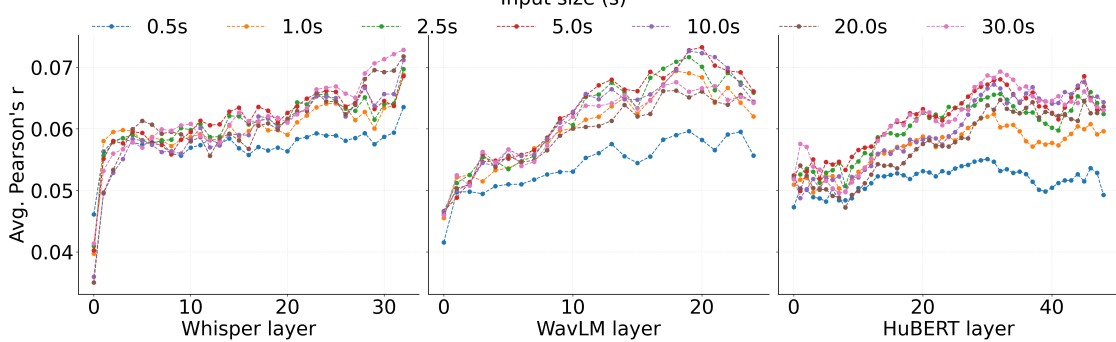

Figure S3: Average correlation between predicted and actual brain responses using embeddings from Whisper, WavLM, and HuBERT across different context lengths. The maximal correlations across layers were obtained with context lengths of 30 s for Whisper, 5 s for WavLM, and 30 s for HuBERT.

### A.5 Variance partitioning with high-level features

In Fig. 2c, the ratio for the phonetic, syntactic, and semantic features was computed using three-way variance partitioning among the SFM representations, speech probability, and the respective feature set (see Fig. S4), in order to control for speech presence information embedded within these higher-level features. For mel-spectrogram and Gabor filter bank features, no such control was applied, as they by definition represent the lowest-level confounds.

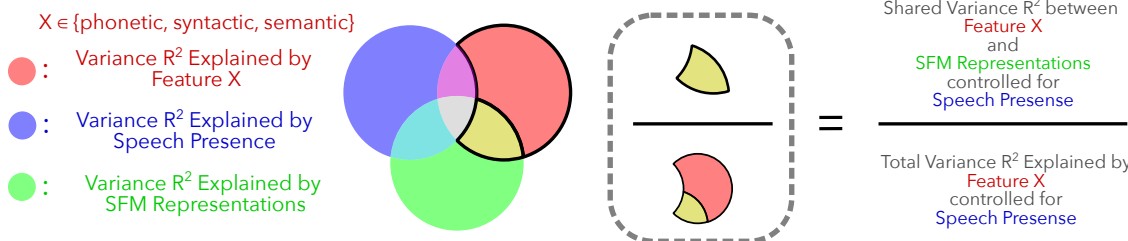

Figure S4: Diagram of the three-way variance partitioning analysis used to quantify the neural variance shared between higher-order features (phonetic, syntactic, and semantic) and SFM representations. Because these higher-order features also encode information about speech presence, we controlled for it explicitly within the partitioning framework.

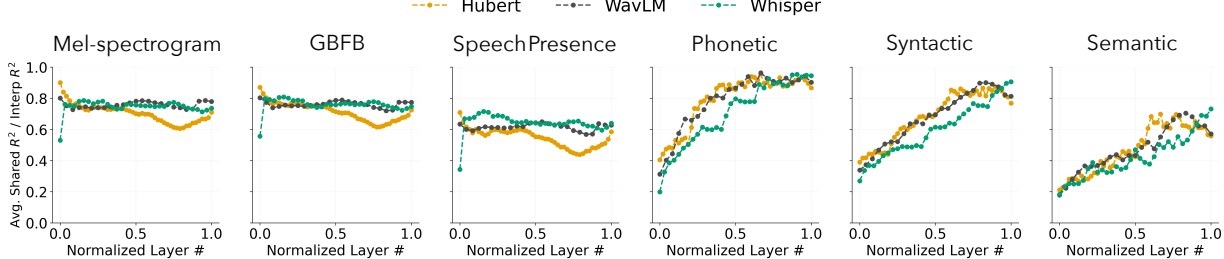

Figure S5: The ratio of neural variance shared between different interpretable feature sets and the SFM representations, relative to the total variance accounted for by each interpretable feature set.

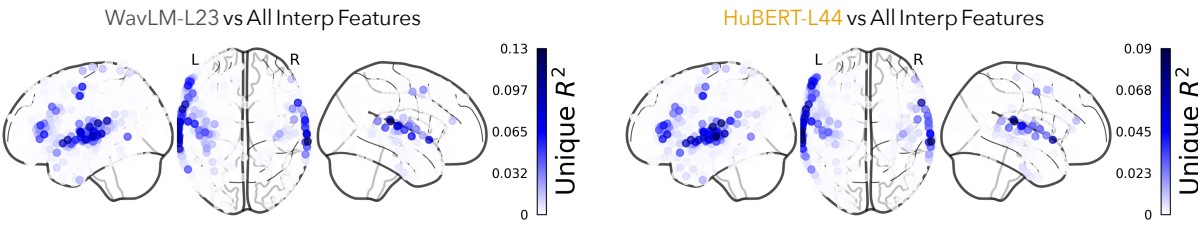

Figure S6: Unique neural variance of WavLM and HuBERT with respect to the combined set of all interpretable features, plotted on a template brain.

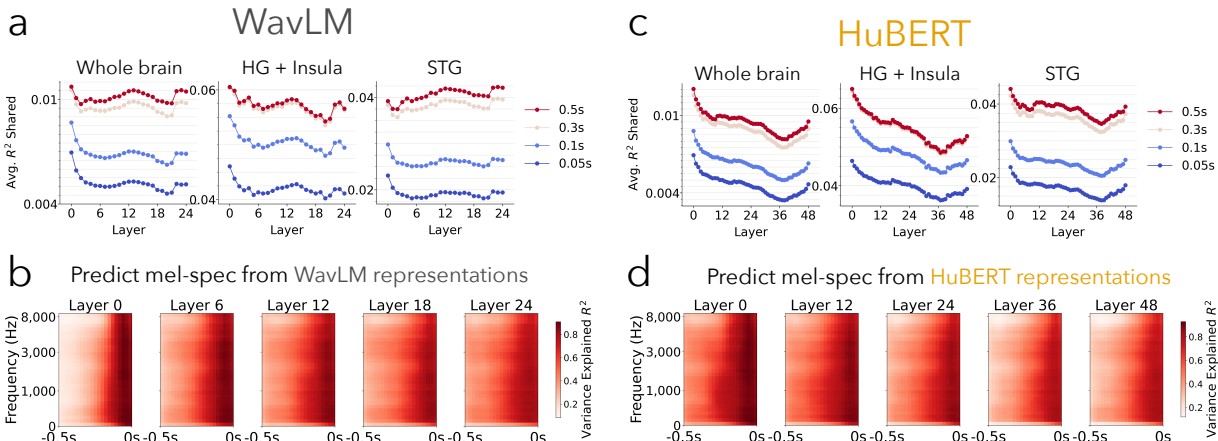

Figure S7: (a) Average neural variance $R^2$ shared between WavLM representations and mel-spectrogram segments of different temporal windows. (b) Variance $R^2$ of the mel-spectrogram $[t - 0.5, t]$s accounted for by the WavLM representations over different layers. (c, d) Corresponding plots for HuBERT.

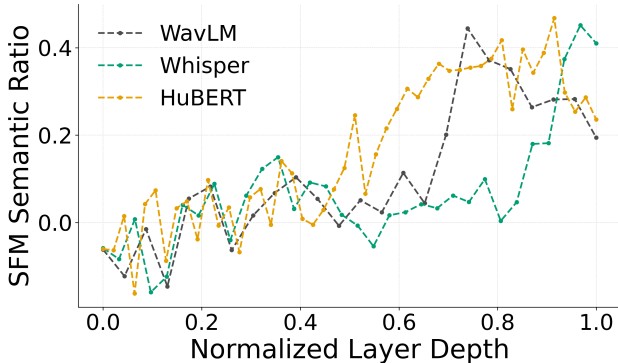

Figure S8: The proportion of unique semantic neural variance accounted for by SFM embeddings over layers, averaged across electrodes in IFG-Tri

Table S1: Summary of the speech foundation models used in this study.

| Model | # Layers | # Dimensions | # Parameters | Training Objective |
|---|---|---|---|---|
| HuBERT X-large | 48 | 1280 | 964M | Masked prediction of *quantised* latent units from iterative k-means clustering of MFCCs |
| WavLM Large | 24 | 1024 | 317M | Same masked-unit objective as HuBERT + utterance-mixing and denoising augmentation |
| Whisper Large | 32 | 1280 | 638M | Supervised multilingual encoder–decoder ASR and speech-to-text translation |

Table S2: List of all questions in question-answering features.

| Question |
| --- |
| 1. Does the sentence describe a personal reflection or thought? |
| 2. Does the sentence contain a proper noun? |
| 3. Does the sentence describe a physical action? |
| 4. Does the sentence describe a personal or social interaction that leads to a change or revelation? |
| 5. Does the sentence involve the mention of a specific object or item? |
| 6. Does the sentence involve a description of physical environment or setting? |
| 7. Does the sentence describe a relationship between people? |
| 8. Does the sentence mention a specific location? |
| 9. Is time mentioned in the input? |
| 10. Is the sentence abstract rather than concrete? |
| 11. Does the sentence express the narrator's opinion or judgment about an event or character? |
| 12. Is the input related to a specific industry or profession? |
| 13. Does the sentence include dialogue? |
| 14. Does the sentence describe a visual experience or scene? |
| 15. Does the input involve planning or organizing? |
| 16. Does the sentence involve spatial reasoning? |
| 17. Does the sentence involve an expression of personal values or beliefs? |
| 18. Does the sentence contain a negation? |
| 19. Does the sentence describe a sensory experience? |
| 20. Does the sentence include technical or specialized terminology? |
| 21. Does the input contain a number? |
| 22. Does the sentence contain a cultural reference? |
| 23. Does the text describe a mode of communication? |
| 24. Does the input include a comparison or metaphor? |
| 25. Does the sentence express a sense of belonging or connection to a place or community? |
| 26. Does the sentence describe a specific sensation or feeling? |
| 27. Does the text include a planning or decision-making process? |
| 28. Does the sentence include a personal anecdote or story? |
| 29. Does the sentence involve a discussion about personal or social values? |
| 30. Does the text describe a journey? |
| 31. Does the input contain a measurement? |
| 32. Does the sentence describe a physical sensation? |
| 33. Does the sentence include a direct speech quotation? |
| 34. Is the sentence reflective, involving self-analysis or introspection? |
| 35. Does the input describe a specific texture or sensation? |
| 36. Is the input about a discovery or realization? |
| 37. Does the sentence include an account of a miscommunication or misunderstanding? |
| 38. Does the sentence include a specific sound or auditory description? |
| 39. Does the sentence use a unique or unusual word? |
| 40. Does the sentence describe a change in a physical or emotional state? |
| 41. Does the sentence describe a moment of relief or resolution of tension? |
| 42. Does the sentence include a conditional clause? |
| 43. Does the sentence reference a specific time or date? |
| 44. Is the sentence conveying the narrator's physical movement or action in detail? |
| 45. Is there mention of a city, country, or geographic feature? |
| 46. Does the sentence involve an unexpected incident or accident? |
| 47. Does the sentence involve a recount of a social or community event? |
| 48. Does the sentence express a philosophical or existential query or observation? |
| 49. Does the story involve a personal project or creation? |
| 50. Is the sentence emotionally positive? |
| 51. Does the sentence describe an activity related to daily life or routine? |
| 52. Does the text include a reference to a past era or time period? |
| 53. Does the input discuss a societal issue or social justice topic? |
| 54. Does the sentence convey a decision or choice made by the narrator? |
| 55. Does the sentence convey a sense of urgency or haste? |
| 56. Is the sentence providing an explanation or rationale? |

