# OpenReview forum: "Interpretable Embeddings of Speech Explain and Enhance the Brain Encoding Performance of Audio Models"
_TMLR — Rejected by TMLR_

### Review · Reviewer_VoVo · 2025-10-17

**Summary Of Contributions:**

The paper studies the relationship between interpretable speech features and SFMs with regards to how well they model speech processing in the human brain. The paper makes a number of findings: first, it finds that SFMs alignment with brain ECoG recordings is largly attributable to interpretable features. Second, the authors find a tradeoff between low-level and high-level features across different layers of the SFMs. Third, the authors find that augmenting SFMs with interpretable features improves correlation with brain recordings. Fourth, the authors find various architectural differences in the types of features best explained by each SFM. Finally, the authors find that SFM representations correlate with brain-relevant semantic information.

**Audience:**

Yes

**Audience Explanation:**

One overarching trend in this field has been to claim that foundation models form good models of perception in the brain. This paper brings some nuance to this claim by showing that much of the predictive power of foundation models is through simple, interpretable features, thus limiting the additional information foundation models provide relative to these interpretable features. Thus, this paper would certainly be of interest to those studying how well large models predict responses in the brain.

**Broader Impact Concerns:**

No broader impact concerns.

**Claims And Evidence:**

No

**Claims Explanation:**

While the paper makes interesting claims, some of them are not fully convincing. First, the paper argues a tradeoff between low-level and high-level features at different layers of SFMs. However, based on Figure 2c, it could be argued that the Mel-spectrogram is mostly flat- there is not significant loss of low-level information at later layers of the SFM. Even looking at figure S4, there doesn't seem to be much of a degradation in low-level features across layers, making the claim of a tradeoff questionable.

Also, the authors claim that augmenting SFMs with interpretable features improves correlation with brain recordings (Figure 3). This is not very surprising itself and needs to be compared against a proper baseline. Augmenting an SFM with *any* features would likely improve correlation. I suggest comparing adding the interpretable features with adding features from another SFM.

Regarding the architectural comparisons made in section 3.3, this does not seem very meaningful or significant given that only 3 SFMs are considered. If the authors wish to make broader claims about architecture in SFMs and their features, I recommend considering more SFMs (at least 5 would be a good start).

**Requested Changes:**

**Critical**
- Explain tradeoff in Figure 2c/Figure S4 (visually, it appears there is not really a tradeoff)
- Add baseline for Figure 3 (compare with addition of a baseline set of features)
- More SFMs in Section 3,.3

---

> ### Author Response · Authors · 2025-10-24
> **Response to Reviewer VoVo**
>
> Thank you for your thoughtful feedback and constructive suggestions. We appreciate the opportunity to clarify our claims and address your concerns. Below, we respond to each of your points in detail.
>
> 1. Explain tradeoff in Figure 2c/Figure S4 (visually, it appears there is not really a tradeoff)
>
> We recognize that when looking solely at Figure 2c/Figure S4, the tradeoff seems modest, as the shared variance with the overall mel-spectrogram feature set does not decrease drastically  for Whisper and WavLM. This puzzling stability is precisely what motivated the deeper analysis in Section 3.3.
>
> Our key finding, presented in Section 3.3, is that this apparent stability masks a more nuanced tradeoff between the precision of acoustic information and the temporal window over which acoustic information is encoded.
>
> - As shown in Figure 4b, the shared neural variance between Whisper and mel-spectrograms with a very short window ([t – 0.05, t]s) consistently decreases across layers. This indicates that later layers progressively lose the ability to represent fine-grained, short-timescale acoustic details.
> - Conversely, the shared variance with longer mel-spectrogram windows ([t – 0.5, t]s) remains stable across layers. This stability occurs because the observed decrease in shared variance with the most recent, short window ([t – 0.05, t]s) is offset by an increase in shared variance with information from the earlier parts ([t - 0.5, t - 0.1]s) of the temporal window.
> - This is further supported by Figure 4c, where we directly predict mel-spectrogram segments from SFM representations. Early layers can reconstruct a short window of spectrogram with high precision, but as we move to later layers, the predictions become temporally elongated but with less precision, reflecting a shift from encoding precise acoustic events to encoding longer-range temporal structure.
>
> Therefore, the tradeoff is not simply a loss of low-level information, but a transformation from precise, local acoustic features to less precise, temporally broad ones. We will revise the manuscript to make this distinction clearer in our initial discussion of Figure 2c and better signpost the more detailed analysis in Section 3.3.
>
> 2. Add baseline for Figure 3 (compare with addition of a baseline set of features)
>
> We thank the reviewer for this feedback. We agree that establishing a proper baseline is essential to support our claim that interpretable features boost the encoding performance of SFMs. Comparing adding the interpretable features with adding features from another SFM would make a good baseline.
>
> Following the reviewer’s suggestion, we conducted an additional experiment in which we computed the test correlation for pairs of models using each model’s best-performing layer (Whisper+HuBERT, Whisper+WavLM, and HuBERT+WavLM; Layer 32 for Whisper and HuBERT, Layer 20 for WavLM). Among these combinations, Whisper+HuBERT achieved the highest average correlation (0.075). By comparison, the highest test correlations for the joint models with interpretable features were 0.085, 0.085, and 0.083 for WavLM, Whisper, and HuBERT, respectively. This difference is statistically significant (p < 0.001, Wilcoxon signed-rank test), further supporting our claim that interpretable features improve encoding model performance.
>
> We once again thank the reviewer for this valuable suggestion.
>
>
> 3. More SFMs in Section 3.3
>
> We acknowledge your concern that making broad, general claims about SFM architectural differences based on only three models is insufficient. The point of Section 3.3, as mentioned above, was not to make a definitive claim about all SFM architectures, but rather to investigate the specific results we observed for Whisper, WavLM, and HuBERT in Section 3.1.
>
> The architectural differences noted in the paper (e.g., convolutional relative positional embeddings in HuBERT vs. fixed sinusoidal embeddings in Whisper) were offered as a plausible hypothesis to explain the distinct behaviors we observed between these specific models. For example, this difference helps explain why HuBERT’s embeddings show a wide temporal receptive field even from the earliest layer, while Whisper’s receptive field starts narrow and expands progressively with layer depth.
>
> We agree that the current phrasing may overstate the generality of these findings. We will revise the manuscript to moderate our claims. We will reframe the discussion to clarify that the architectural differences are presented as a plausible hypothesis to explain the specific phenomena we observed with three models, rather than as a broad, generalizable claim about all SFMs.

---

### Review · Reviewer_5ZwM · 2025-11-06

**Summary Of Contributions:**

This paper investigates how Speech Foundation Models (SFMs: Whisper, HuBERT, WavLM) align with brain activity recorded via ECoG during naturalistic listening. By systematically comparing SFM embeddings with six "interpretable" acoustic/linguistic features (mel-spectrograms, Gabor filters, phonetic, syntactic, semantic, etc.), the authors perform variance partitioning to assess how much of the SFM–brain alignment can be explained by these interpretable features.

Key strengths

- Well-executed variance-partitioning pipeline that compares SFMs and multiple interpretable feature families on ECoG data.
- Uses appropriate encoding methods (banded ridge) and performs layerwise analysis.
- Provides a useful, practical recommendation (augment SFM embeddings with interpretable features) with demonstrated improvements in prediction.

Key weaknesses

- Limited generalizability: nine clinical participants and a single 30-minute stimulus restrict how broadly results can be interpreted.
- The GPT-4 QA semantic embeddings are themselves produced by black-box LLMs, weakening claims of interpretability.
- Important methodological details and potential confounds are under-specified (downsampling, cross-feature correlations, noise ceiling / explainable variance, statistical corrections).
- Some claims verge on usage of causal language.

**Additional Comments:**

Minor corrections:

Section 2.1: change “Datasets” → “Dataset.”

Section 2.2: change “Interpretable features” → “Interpretable Features” (capital F).

**Audience:**

Yes

**Audience Explanation:**

The work is highly relevant to audiences at the intersection of machine learning, cognitive neuroscience, and representation interpretability. It offers a valuable methodological bridge between self-supervised audio modeling and neural encoding, a growing area of interest in TMLR. However, the novelty is more incremental than transformative, as similar representational decomposition analyses have been performed in prior LLM–brain studies. The main novelty lies in extending this to SFMs.

**Broader Impact Concerns:**

Can briefly mention potential misuse (speech/brain decoding) as well.

**Claims And Evidence:**

No

**Claims Explanation:**

Partially. Some concerns:

Limited generalizability:
The small sample size (nine ECoG participants) and reliance on a single 30-minute podcast stimulus constrain external validity and increase susceptibility to overfitting or stimulus-specific correlations.

Semantic feature reproducibility:
The use of GPT-4-derived QA semantic features limit interpretability and reproducibility. These features are not truly “hand-crafted” and may encode hidden confounds from the LLM itself.

Variance partitioning caveats:
High intercorrelations among features may bias the estimation of “unique” variance. Features in variance partitioning are assumed to be orthogonal; this needs to be demonstrated.

Lack of noise ceilings or explainable variance:
Without a reported noise ceiling or measure of explainable neural variance, it is unclear whether observed R² values approach the data’s reliability limits or reflect modest fits.

Overstated interpretational language:
Claims of mechanistic explanation (in the intro) or causal insight should be softened. The findings describe statistical alignment between models and neural data, not mechanistic accounts of neural computation.

**Requested Changes:**

Introduction

- Remove the phrase “mechanistic understanding” — the analyses are not mechanistic. Or explicitly state what you mean by this.

- Add supporting references for every sentence that lists a claim.   For example reference is missing for "early layers encode acoustic/phonetic features while later layers capture higher-level linguistic information."

- Rephrase the central question into a simpler, reader-friendly sentence.

- Clearly define or explicitly state what is meant by “systematic trade-off”.

Methods

- Clarify how features were downsampled to 4 Hz. what interpolation was used, and were all features processed identically?

- Syntactic features: Why were full parse trees or richer syntactic embeddings not used? Current features (POS, dependencies) may be too weak.

- Semantic features: Explain why GPT-4 QA embeddings were chosen despite limited reproducibility; consider simpler interpretable alternatives such as WordNet or English1000 or Fastext.

- Provide the approximate number of words in a 0.5 s window to interpret temporal alignment.

- Add a noise ceiling or explainable variance estimate (e.g., split-half reliability).

- Clarify that not all models were fit using banded ridge regressio - only when there are multiple feature spaces.

Results / Figures

- Use consistent terminology: replace “explain” with “account for shared variance” in figure captions and text.

- Indicate whether scores are normalized (per electrode, subject, or brain region) and include error bars where possible.  Error bars can be added by bootstrapping across different subsets of participants.

-  To strengthen the generalisability of the results can the analysis be performed on another dataset (eeg/meg)? This can be put in the supplementary.

Discussion

- Is it 40% ? Please clarify. Seems “4%” in Fig. 5D.

- Moderate the causal phrasing in the results.

-  Strengthen the discussions by explicitly explaining why the conclusions differe despite similar questions in the studies mentioned.

- Would the dimensionality argument for Whisper vs. mel-spectrogram hold, if both are PCA-reduced?

---

> ### Author Response · Authors · 2025-11-26
> **Response to Reviewer 5ZwM**
>
> We thank the reviewer for their thorough and constructive feedback. We have addressed each point below and believe the suggested changes have significantly improved the manuscript.
>
> ## Requested Changes:
>
> ### Introduction
>
> **Remove the phrase "mechanistic understanding" — the analyses are not mechanistic. Or explicitly state what you mean by this.**
>
> We agree that our analyses, being correlational, do not support mechanistic claims. We have removed this phrase from the manuscript.
>
> **Add supporting references for every sentence that lists a claim. For example reference is missing for "early layers encode acoustic/phonetic features while later layers capture higher-level linguistic information."**
>
> We have added appropriate citations for this claim.
>
> - Pasad, A., Chou, J. C., & Livescu, K. (2021, December). Layer-wise analysis of a self-supervised speech representation model. In 2021 IEEE Automatic Speech Recognition and Understanding Workshop (ASRU) (pp. 914-921). IEEE.
> - Pasad, A., Shi, B., & Livescu, K. (2023, June). Comparative layer-wise analysis of self-supervised speech models. In ICASSP 2023-2023 IEEE International Conference on Acoustics, Speech and Signal Processing (ICASSP) (pp. 1-5). IEEE.
> - Pasad, A., Chien, C. M., Settle, S., & Livescu, K. (2024). What do self-supervised speech models know about words?. Transactions of the Association for Computational Linguistics, 12, 372-391.
>
> **Rephrase the central question into a simpler, reader-friendly sentence. Clearly define or explicitly state what is meant by "systematic trade-off".**
>
> We agree that a simpler framing of the central question and an explicit definition of the "systematic trade-off" will improve the manuscript's clarity.
>
> Our central questions can be stated more simply as:
> - How much of SFM's brain predictive power is accounted for by interpretable features?
> - As we move through SFM's layers, do they merely accumulate information that is useful for neural encoding, or do they exhibit more specific, feature-dependent patterns?
>
> By a "systematic trade-off", we refer to the layer-by-layer pattern in which SFM shows:
> - loss of acoustic features relevant for neural processing.
> - a corresponding increase in brain-relevant higher-level information (phonetic, syntactic, and semantic features)
>
> We have updated the manuscript to reflect these points.
>
> ### Methods
>
> **Clarify how features were downsampled to 4 Hz. what interpolation was used, and were all features processed identically?**
>
> Below, we describe the procedure for each feature type and specify whether interpolation was used. In all cases, we produced one feature vector every 0.25 s (i.e., 4 Hz), but the method for obtaining each vector differed according to the feature representation.
>
> - **Melspec**: To compute the mel-spectrogram feature at time $t$, we input the waveform from $[t - 0.5 \text{ s}, t]$ into the standard mel filterbank, producing a 2D spectrogram with 32 mel bins $\times$ 21 time frames, corresponding to the most recent 0.5 s. This matrix was flattened to a 672-dimensional vector at each 4 Hz time point. PCA was then applied to the full set of these flattened vectors.
> - **GBFB**: GBFB features were originally computed at 100 Hz (455 dimensions). To convert these to 4 Hz, we resampled the time series to 0.25-s intervals using `scipy.signal.resample`. Thus, it uses band-limited sinc interpolation.
> - **Speech probability**: At each 4 Hz time point, the audio segment $[t - 0.5 \text{ s}, t]$ was passed to the audio tagging model, and we used the model's output probability as a single scalar feature. No interpolation was applied.
> - **Phonetic, Syntactic, and Semantic features**: For linguistic features, we identified all phonemes or words whose offset fell within the window $[t - 0.5 \text{ s}, t]$. Each phoneme/word was represented by its corresponding feature embedding. The final 4 Hz feature vector at time $t$ was obtained by summing the embeddings of all items in that window. Again, no interpolation was used.

---

> > ### Author Response · Authors · 2025-11-26
> >
> > **Syntactic features: Why were full parse trees or richer syntactic embeddings not used? Current features (POS, dependencies) may be too weak.**
> >
> > While richer syntactic embeddings like full parse trees are valuable, we strategically selected part-of-speech (POS) tags and dependency labels for two main reasons.
> >
> > First, we aimed to ground our analysis in syntactic representations that are well-established and have been extensively validated as contributing to brain encoding models. A large body of literature (See the list below) has successfully used both POS tags and dependency structures to encode/decode and explain neural activity related to syntax. By using these features, we can more directly compare the representations learned by SFMs to known neural correlates of syntax.
> >
> > While more complex features are an exciting avenue for research (e.g., Oota et al., 2023), our primary goal was to investigate how Speech Foundation Models (SFMs) encode these known, brain-relevant syntactic constructs across their layers. As shown in Figure 2C of our manuscript, these features were highly effective for this purpose, clearly revealing a gradual increase in syntactic encoding in deeper SFM layers. This demonstrates that the chosen features were sufficiently powerful to test our core hypotheses.
> >
> > - **POS**: Murphy, A., Bohnet, B., McDonald, R., & Noppeney, U. (2022, May). Decoding part-of-speech from human EEG signals. In Proceedings of the 60th Annual Meeting of the Association for Computational Linguistics (Volume 1: Long Papers) (pp. 2201-2210).
> > - **POS**: Vigliocco, G., Vinson, D. P., Druks, J., Barber, H., & Cappa, S. F. (2011). Nouns and verbs in the brain: A review of behavioural, electrophysiological, neuropsychological and imaging studies. Neuroscience & Biobehavioral Reviews, 35(3), 407-426.
> > - **POS**: Wehbe, L., Murphy, B., Talukdar, P., Fyshe, A., Ramdas, A., & Mitchell, T. (2014). Simultaneously uncovering the patterns of brain regions involved in different story reading subprocesses. PloS one, 9(11), e112575.
> > - **Dependency**: Lopopolo, A., Van den Bosch, A., Petersson, K. M., & Willems, R. M. (2021). Distinguishing syntactic operations in the brain: Dependency and phrase-structure parsing. Neurobiology of Language, 2(1), 152-175.
> > - **POS and Dependency**: Wehbe, L., Murphy, B., Talukdar, P., Fyshe, A., Ramdas, A., & Mitchell, T. (2014). Simultaneously uncovering the patterns of brain regions involved in different story reading subprocesses. PloS one, 9(11), e112575.
> > - **POS and Dependency**: Caucheteux, C., Gramfort, A., & King, J. R. (2021, July). Disentangling syntax and semantics in the brain with deep networks. In International conference on machine learning (pp. 1336-1348). PMLR.
> > - **POS and Dependency**: Reddy, A. J., & Wehbe, L. (2021). Can fMRI reveal the representation of syntactic structure in the brain?. Advances in neural information processing systems, 34, 9843-9856.
> > - **Constituency and Dependency parsing**: Oota, S. R., Marreddy, M., Gupta, M., & Bapi, R. (2023, July). How does the brain process syntactic structure while listening?. In Findings of the Association for Computational Linguistics: ACL 2023 (pp. 6624-6647).

---

> > > ### Author Response · Authors · 2025-11-26
> > >
> > > **Semantic features: Explain why GPT-4 QA embeddings were chosen despite limited reproducibility; consider simpler interpretable alternatives such as WordNet or English1000 or Fastext.**
> > >
> > > We chose QA embeddings over other interpretable semantic features like WordNet or English1000 because of their two distinct advantages.
> > > - The prior work by Singh et al. (2025) showed that QA features constructed with as few as 35 questions significantly outperformed English1000 in predicting ECoG responses, tested on the same Podcast dataset used in this paper. Additionally, the representation is more compact, which makes it suitable for ECoG signals that have a limited number of samples.
> > > - QA embeddings are arguably more interpretable than the alternatives. In QA embeddings, each dimension corresponds to a binary (0/1) value of a specific, human-readable question (e.g., "Is the sentence abstract rather than concrete?"). This is more interpretable compared to the alternative embeddings, where features exist in dense, latent dimensions (some arbitrary statistical measure of word relations) that are difficult to interpret.
> > >
> > > Regarding the valid concern about reproducibility with proprietary models like GPT-4, we acknowledge this limitation. However, we have ensured that our methodology is fully transparent and reproducible. The complete list of 56 questions used to generate the embeddings is provided in our manuscript (Table S2), following the open methodology of Benara et al. (2024). The answers to these questions can be found in CSV files attached as supplementary material to this paper. This ensures that the results in this paper are reproducible.
> > >
> > > - Benara, V., Singh, C., Morris, J. X., Antonello, R., Stoica, I., Huth, A. G., & Gao, J. (2024). Crafting interpretable embeddings by asking llms questions. arXiv preprint arXiv:2405.16714.
> > > - Singh, C., Antonello, R. J., Guo, S., Mischler, G., Gao, J., Mesgarani, N., & Huth, A. G. (2025). Evaluating scientific theories as predictive models in language neuroscience. bioRxiv, 2025-08.
> > >
> > > **Provide the approximate number of words in a 0.5s window to interpret temporal alignment.**
> > >
> > > To aid in interpreting the temporal alignment of our features, we quantified the word density within the 0.5-second integration windows used in our model. Specifically, for each time point $t$ (sampled at 4 Hz), we counted the number of words whose offset timestamps fell within the interval $[t-0.5\text{s},t]$.
> > >
> > > Over the 30-minute duration (1800 seconds, resulting in $N=7200$ samples), the distribution of word counts per window is as follows:
> > >
> > > | # of Words | Count | Percentage |
> > > |------------|-------|------------|
> > > | 0 | 1,389 | 19.3% |
> > > | 1 | 2,679 | 37.2% |
> > > | 2 | 2,026 | 28.1% |
> > > | 3 | 907 | 12.6% |
> > > | 4 | 175 | 2.4% |
> > > | 5 | 24 | 0.3% |
> > >
> > > Notably, the rare instances of 5 words in a 0.5-second window (0.3%) are attributable to moments of overlapping speech between multiple speakers. We have add this information as a supplementary figure.
> > >
> > > **Add a noise ceiling or explainable variance estimate (e.g., split-half reliability).**
> > >
> > > We thank the reviewer for this suggestion. We agree that a noise ceiling provides a valuable benchmark. However, calculating a true noise ceiling (e.g., via split-half reliability) requires repeated presentations of the same stimuli. As our study utilizes a naturalistic ECoG dataset without trial repetitions, a direct noise ceiling calculation is not feasible.
> > >
> > > However, to address the need for variance estimation and ensure our results are not driven by noise, we have added error bars (95% confidence intervals) to all performance metrics. These were calculated by bootstrapping across electrodes. While this does not define the absolute upper bound of explainable variance, it confirms that the relative performance differences between models, the primary focus of our claims, are robust and statistically reliable.
> > >
> > > **Clarify that not all models were fit using banded ridge regression - only when there are multiple feature spaces.**
> > >
> > > We agree with this suggestion and have updated the manuscript accordingly.

---

> > > > ### Author Response · Authors · 2025-11-26
> > > >
> > > > ### Results / Figures
> > > >
> > > > **Use consistent terminology: replace "explain" with "account for shared variance" in figure captions and text.**
> > > >
> > > > We agree with this suggestion and have updated the manuscript accordingly.
> > > >
> > > > **Indicate whether scores are normalized (per electrode, subject, or brain region) and include error bars where possible. Error bars can be added by bootstrapping across different subsets of participants.**
> > > >
> > > > We thank the reviewer for this important point. We agree that providing a measure of variance is critical for interpreting results. The reviewer suggests bootstrapping across participants, a technique that is indeed suitable for many neuroimaging modalities where recording locations are consistent across individuals (e.g., EEG/fMRI). However, for ECoG data, this approach can be misleading.
> > > >
> > > > Because ECoG electrode placement on the brain is determined by clinical needs, coverage is sparse and uneven across subjects. A brain region critical for encoding of specific features (e.g., Inferior Frontal Gyrus for semantic processing) might not be covered by all participants. Consequently, bootstrapping by subject would create unstable variance estimates, artificially inflating the error bars.
> > > >
> > > > We have conducted bootstrapping across all electrodes to estimate 95% confidence interval and have updated the figures accordingly.
> > > >
> > > > **To strengthen the generalisability of the results can the analysis be performed on another dataset (eeg/meg)? This can be put in the supplementary.**
> > > >
> > > > We thank the reviewer for raising the important point of generalizability. We specifically chose ECoG for this study, as its high signal-to-noise ratio (SNR) and fine spatial resolution were essential for the analyses at the core of our paper.
> > > >
> > > > Our primary analytical tool, variance partitioning, aims to isolate the unique neural variance explained by different feature sets. These unique contributions, while statistically significant, can be small. With the lower SNR inherent in EEG/MEG, these components would likely be obscured by noise, making it difficult to conduct variance-partitioning with the same fidelity.
> > > >
> > > > Therefore, while we agree that testing our findings on other datasets is a valuable direction for future research, ECoG provides the most appropriate and powerful modality for the fine-grained representational analysis we present. We have updated the limitations section to make this choice clear.

---

> > > > > ### Author Response · Authors · 2025-11-26
> > > > >
> > > > > ### Discussion
> > > > >
> > > > > **Is it 40% ? Please clarify. Seems "4%" in Fig. 5D.**
> > > > >
> > > > > We agree that the current wording and figure are ambiguous, and we appreciate the opportunity to clarify.
> > > > >
> > > > > The "40%" figure does not refer to the absolute $R^2$ values shown in Figure 5D. Instead, it represents the proportion of unique semantic variance that is also accounted for by the SFM representations. In terms of the Venn diagram in Figure 5A, this corresponds to the ratio Var(Yellow) / Var(Yellow + Red).
> > > > >
> > > > > To make the connection to the figures explicit:
> > > > > - Var(Yellow), plotted in Figure 5E, is approximately 0.0026 across models.
> > > > > - Var(Yellow + Red), shown in the boxplot in Figure 5D, has a mean of approximately 0.0058.
> > > > >
> > > > > Using these values, the ratio is $0.0026 / 0.0058 = 44.8$%, which underlies our statement that "about 40%" of the unique semantic variance is captured by the SFM representations.
> > > > >
> > > > > We will update the Figure 5E to convert the absolute value to the ratio so that the interpretation of the figure is more intuitive.
> > > > >
> > > > > **Moderate the causal phrasing in the results.**
> > > > >
> > > > > We agree with this suggestion and have updated the manuscript accordingly.
> > > > >
> > > > > **Strengthen the discussions by explicitly explaining why the conclusions differ despite similar questions in the studies mentioned.**
> > > > >
> > > > > We thank the reviewer for raising this crucial point. We agree that reconciling our findings, specifically that SFMs do encode brain-relevant semantics, with the conclusions of Oota et al. (2024) and Choi et al. (2024) is essential for placing our contributions in the correct context.
> > > > >
> > > > > Regarding Oota et al. (2024), we believe the apparent contradiction stems primarily from differences in model scale and layer selection.
> > > > >
> > > > > Oota et al. (2024) conclude that the alignment of SFMs with language regions is not due to semantics. However, their analysis relied on Whisper-small and wav2vec2.0-base. As demonstrated in our Figure 5e (right panel), the capacity of SFMs to encode brain-relevant semantic information is highly sensitive to model size. We observe a clear scaling law where smaller variants (like Whisper-small) encode significantly less unique semantic variance compared to the larger variants used in our primary analysis (Whisper-large). By restricting their analysis to smaller models, Oota et al. likely underestimated the semantic capabilities inherent in the architecture at scale.
> > > > >
> > > > > Furthermore, Oota et al. extracted embeddings from intermediate layers of Whisper-small. Our results, shown in Figure 2c, indicate that semantic alignment in Whisper peaks in the final layers.
> > > > >
> > > > > Regarding Choi et al. (2024), their work suggests that SFMs are more phonetic than semantic, rather than devoid of semantics entirely. This actually aligns with our findings in Figure 2c, where we show that while high-level semantic encoding increases with depth, lower-level features (like phonetics) account for a large portion of the variance throughout. Our variance partitioning (Fig 5c) confirms that while SFMs capture meaningful semantics, they only account for approximately 40% of the unique variance explained by our interpretable semantic features. Thus, we agree with the spirit of Choi et al. that SFM representations are not purely or perfectly semantic, but we provide strong evidence that semantic encoding is present and statistically significant in large-scale models.
> > > > >
> > > > > We have updated the manuscript to explicitly state this point.
> > > > >
> > > > > **Would the dimensionality argument for Whisper vs. mel-spectrogram hold, if both are PCA-reduced?**
> > > > >
> > > > > Our analysis confirms that the dimensionality argument remains valid even when both representations are PCA-reduced.
> > > > >
> > > > > We performed a direct comparison: to explain 95% of the variance in a 20 ms audio window, a PCA-reduced mel-spectrogram was condensed to just 7 dimensions. In contrast, embeddings from Whisper's 0th layer required 171 dimensions to capture the same variance threshold, a 24-fold difference.
> > > > >
> > > > > This is likely because while Whisper's input is a mel-spectrogram, it immediately applies normalization, two convolutional layers with GeLU activations, and adds sinusoidal positional embeddings. These non-linear operations create a higher-dimensional representation that is not as efficiently reduced by PCA.
> > > > >
> > > > > Given that this expanded representation from Whisper's 0th layer explains no unique neural variance compared to our interpretable features, it underscores our conclusion that the SFM begins with a less compact and less efficient encoding of basic acoustic information.

---

### Review · Reviewer_az4Y · 2025-11-18

**Summary Of Contributions:**

The work proposes a set of interpretable features of speech to study and enhance speech foundation models’  (SFMs) alignment with the brain. It carries out extensive variance partitioning analyses for ECoG responses to naturalistic speech stimuli to provide insight into the representational alignment of SFMs and how it changes across layers and model families. Overall, the paper is very well-written, and the motivation is clear. The work, however, doesn’t provide many novel insights or methods, and some results need to be reported more rigorously (see weaknesses).

## **Strengths**

1. The motivation of the work is well-timed and well-posed as it’s relevant to the recent directions of interpretable embeddings for brain alignment.
2. The paper is very well-written, the method is easy to follow, and the results are reported in a clear fashion.
3. Clear and deep analysis for the reported results, providing concise insights and clarifying caveats and limitations for each experiment.
4. The results highlighting the existence of brain-relevant semantics and the acoustic temporal window tradeoff are useful and highly relevant to the literature in this field.
5. The claims made in the paper are supported by sufficient analyses that are mostly rigorous, and there is no reason to suspect any kind of cherry picking or attempts to oversell the findings.
6. The authors seem to be well aware of the relevant recent literature and did a (mostly) good job discussing it when needed.
7. The detailed explanation of all the methods is sufficient to reproduce all the results, especially since the used dataset is publicly available.


## Weaknesses
See below.

**Additional Comments:**

**Minor:** for figs 5 and 2e, it might be helpful to report the percentage of the variance rather than the absolute number as done in fig. 2 b.

**Audience:**

Yes

**Audience Explanation:**

The findings of this paper are relevant for people working on speech models, brain alignment, computational neuroscience, and linguistics. Moreover, the results highlighting the existence of brain-relevant semantics and the acoustic temporal window tradeoff are new and insightful, making them highly relevant to this field.

**Broader Impact Concerns:**

I don't have concerns about that as the authors reported openly using LLMs to help with the writing and also reported well the limitations and broader impact of the study and dataset used.

**Claims And Evidence:**

No

**Claims Explanation:**

While the analyses in the paper are mostly rigorous, there are multiple points that need further clarification and/or additional results:

1. **Novelty is limited as the methods and many of the findings have been established before in previous SFM/brain work, especially the** usage of low-level features and LLM QA features in brain encoding. However, the insights provided are still valuable, provided that the concerns below are addressed.
2. The reported results in Figure 3 don’t provide a strong baseline or analyze the effect of each individual feature, making it unclear what features are contributing to this improvement and how it compares to other baselines. Would it be possible to a) analyze this improvement with each individual feature and b) add a simple baseline like combining representations from early and late layers to account for high and low-level features?
3. Although the paper does a good job of covering the literature, the discussion of some relevant work is missing from the introduction. For instance, the claim that “The existing work on SFM-brain alignment has largely focused on SFM’s predictive success….” is not totally accurate, as previous work (like Oota et al., 2024) discussed what low-level features drive representations of speech models and even argued that they lack brain-relevant semantics. A better contextualization and discussion of such work is important to add there, especially since the authors provide different arguments.
4. It’s unclear why only the average was reported in the results across the paper, with no mention of the standard error across participants. Providing STE is essential for understanding the consistency of the results. Can the authors comment on why they didn’t report it anywhere?
5. While the results of Fig.5 are insightful, it’s still unclear how this shared variance changes across model layers.
6. It is not explicitly mentioned whether PCA was fit only on training data per fold (as it should be) or on the full dataset before cross-validation.

**Requested Changes:**

**In addition** to the questions mentioned above, would the author comment further on how to reconcile the findings from Oota et al., 2024 and Choi et al., 2024 with the results from this work? I believe a clear argument and discussion of this needs to be included in the paper as well.

---

> ### Author Response · Authors · 2025-11-26
>
> We thank the reviewer for their positive assessment and for the many constructive comments that helped improve this paper.
>
> **The reported results in Figure 3 don't provide a strong baseline or analyze the effect of each individual feature, making it unclear what features are contributing to this improvement and how it compares to other baselines. Would it be possible to a) analyze this improvement with each individual feature and b) add a simple baseline like combining representations from early and late layers to account for high and low-level features?**
>
> a) We agree that determining which features explain neural variance beyond what SFM embeddings capture is essential for interpretation. We believe this question can be addressed by re-examining Figure 2c. The performance gains shown in Figure 3 arise from the portion of neural variance that interpretable features capture but SFM embeddings do not. This difference corresponds directly to the inverse of the values in Figure 2s; that is, $1 - (R^2_{\text{shared}} / R^2_{\text{interp}})$. For example, phonetic features are not well represented in SFM embeddings in the early layers, so they contribute substantial unique variance in these layers. However, in later layers, SFMs capture nearly all phonetic information, meaning these features no longer provide unique variance. Thus, semantic and lower-level acoustic features are the primary drivers of the observed improvements, whereas phonetic and syntactic features are largely accounted for by SFM embeddings.
>
> We will update the manuscript to clarify this connection.
>
> b) We also agree that having a baseline would be crucial to establish the validity of the improvement. For this, we have added two kinds of baselines: 1. Encoding model using each SSM's 0th layer and its best-performing layer (Layer 32 for Whisper and HuBERT, Layer 20 for WavLM). 2. Encoding model using best-performing layers of two SSMs (Whisper+HuBERT, Whisper+WavLM, and HuBERT+WavLM).
>
> | Model | Baseline 0 (1 Layer, 1 SFM) | Baseline 1 (Early+Late) | Baseline 2 (Multi-Model) | Our Method (SFM+Interp) |
> |-------|----------------------------|-------------------------|--------------------------|-------------------------|
> | Whisper | 0.073 | 0.075 | 0.075 | 0.085 |
> | HuBERT | 0.069 | 0.072 | 0.075 | 0.083 |
> | WavLM | 0.073 | 0.074 | 0.074 | 0.085 |
>
> As can be observed from this table, while the baselines improve encoding performance, the joint model of SSM embeddings and interpretable features outperforms them with statistical significance ($p < 0.001$, Wilcoxon signed-rank test).
>
> **Although the paper does a good job of covering the literature, the discussion of some relevant work is missing from the introduction. For instance, the claim that "The existing work on SFM-brain alignment has largely focused on SFM's predictive success…." is not totally accurate, as previous work (like Oota et al., 2024) discussed what low-level features drive representations of speech models and even argued that they lack brain-relevant semantics. A better contextualization and discussion of such work is important to add there, especially since the authors provide different arguments.**
>
> Thank you for pointing this out. We acknowledge that recent work (Oota et al., 2024; Anderson et al., 2024) has begun to address this question. We have modified the manuscript to highlight that while there are a few studies that have attempted to address this question, their results are conflicting and inconclusive. For example, while Oota et al. (2024) argue that SFM lacks brain-relevant semantic information, Anderson et al. (2024) provide evidence suggesting the presence of brain-relevant lexical encoding.

---

> > ### Author Response · Authors · 2025-11-26
> >
> > **It's unclear why only the average was reported in the results across the paper, with no mention of the standard error across participants. Providing STE is essential for understanding the consistency of the results. Can the authors comment on why they didn't report it anywhere?**
> >
> > We thank the reviewer for this important point. We agree that providing a measure of variance is critical for interpreting results.
> >
> > We initially did not include STE across participants as we believed it is not well-suited for our datasets. Because ECoG electrode placement on the brain is determined by clinical needs, coverage is sparse and uneven across subjects. A brain region critical for encoding specific features (e.g., Inferior Frontal Gyrus for semantic processing) might not be covered by all participants. Consequently, calculating STE across participants for feature-specific metrics (Figure 2c, for example) would create unstable variance estimates, artificially inflating the error bars.
> >
> > We have conducted bootstrapping across all electrodes to estimate 95% confidence interval and have updated the figures accordingly.
> >
> > **While the results of Fig.5 are insightful, it's still unclear how this shared variance changes across model layers.**
> >
> > The shared variance in Fig. 5 increases with layers, following the same trend as the plot in Figure 2c for semantic features. We will add this plot as a supplementary figure and also explicitly mention the trend in the manuscript.
> >
> > **It is not explicitly mentioned whether PCA was fit only on training data per fold (as it should be) or on the full dataset before cross-validation.**
> >
> > PCA was fit on training data per fold and then applied to the test set. We will update the manuscript to explicitly state this.
> >
> > **In addition to the questions mentioned above, would the author comment further on how to reconcile the findings from Oota et al., 2024 and Choi et al., 2024 with the results from this work? I believe a clear argument and discussion of this needs to be included in the paper as well.**
> >
> > We thank the reviewer for raising this crucial point. We agree that reconciling our findings, specifically that SFMs do encode brain-relevant semantics, with the conclusions of Oota et al. (2024) and Choi et al. (2024) is essential for placing our contributions in the correct context.
> >
> > Regarding Oota et al. (2024), we believe the apparent contradiction stems primarily from differences in model scale and layer selection.
> >
> > Oota et al. (2024) conclude that the alignment of SFMs with language regions is not due to semantics. However, their analysis relied on Whisper-small and wav2vec2.0-base. As demonstrated in our Figure 5e (right panel), the capacity of SFMs to encode brain-relevant semantic information is highly sensitive to model size. We observe a clear scaling law where smaller variants (like Whisper-small) encode significantly less unique semantic variance compared to the larger variants used in our primary analysis (Whisper-large). By restricting their analysis to smaller models, Oota et al. likely underestimated the semantic capabilities inherent in the architecture at scale.
> >
> > Furthermore, Oota et al. extracted embeddings from intermediate layers of Whisper-small. Our results, shown in Figure 2c, indicate that semantic alignment in Whisper peaks in the final layers.
> >
> > Regarding Choi et al. (2024), their work suggests that SFMs are more phonetic than semantic, rather than devoid of semantics entirely. This actually aligns with our findings in Figure 2c, where we show that while high-level semantic encoding increases with depth, lower-level features (like phonetics) account for a large portion of the variance throughout. Our variance partitioning (Fig 5c) confirms that while SFMs capture meaningful semantics, they only account for approximately 40% of the unique variance explained by our interpretable semantic features. Thus, we agree with the spirit of Choi et al. that SFM representations are not purely or perfectly semantic, but we provide strong evidence that semantic encoding is present and statistically significant in large-scale models.
> >
> > We have updated the manuscript to explicitly state this point.
> >
> > **Minor: for figs 5 and 2e, it might be helpful to report the percentage of the variance rather than the absolute number as done in fig. 2 b.**
> >
> > We agree that for Figure 5e, reporting the percentage makes the interpretation significantly clearer, as it highlights the proportion of the semantic feature space that the SFMs are able to capture. We have updated Figure 5e.
> >
> > For Figures 2d and 2e, we have opted to retain the absolute variance values. If we were to report percentages here (i.e., unique variance divided by total SFM variance), this metric would be confounded by the fact that the total variance explained by SFMs increases over layers. Here, we aim to quantify the absolute amount of unique variance by SFMs.

---

### Author Response · Authors · 2025-11-26
**Overall Response**

We thank the reviewers for their thoughtful and constructive feedback. We were encouraged by the positive reception of our work and the recognition of the importance of understanding how Speech Foundation Models (SFMs) align with neural processing. The reviewers' comments have been instrumental in refining our analysis and strengthening our claims.

Based on your suggestions, we have made revisions to the manuscript. Below, we summarize the major changes:

## 1. Strengthened Baselines for Figure 3

A shared concern was the need for stronger baselines to validate the claim that interpretable features improve encoding performance beyond SFM embeddings alone. We have introduced two baselines:

* **Early + Late Layer Baseline**: Combining the 0th layer with the best-performing layer of the same model.
* **Multi-Model Baseline**: Combining the best-performing layers from two different SFMs (e.g., Whisper + HuBERT).

Our analysis shows that the joint model of SFM embeddings plus interpretable features significantly outperforms both baselines ($p < 0.001$, Wilcoxon signed rank test), confirming that interpretable features capture unique neural variance not represented by current SFMs.

## 2. Contextualization with Recent Literature

We have expanded our discussion to explicitly reconcile our findings with recent work by Oota et al. (2024) and Choi et al. (2024). We clarify that apparent contradictions regarding semantic encoding arise primarily from differences in model scale and layer selection.

* We demonstrate a clear scaling law (Fig. 5E), showing that larger models (e.g., Whisper-large, used in our study) capture significantly more semantic variance than the smaller variants used in prior negative results. Furthermore, Oota et al. extracted embeddings from intermediate layers of Whisper-small. Our results, shown in Figure 2c, indicate that semantic alignment in Whisper peaks in the final layers.
* We clarify that while SFMs do encode brain-relevant semantics, they account for approximately 44% of the unique variance explained by interpretable semantic features, aligning with the nuance that SFMs are not perfectly semantic but do possess significant semantic alignment.

## 3. Statistical Robustness and Error Bars

To ensure our results are not driven by noise, we have added 95% confidence intervals to all performance metrics (Figures 2, 3, and 5). These were calculated by bootstrapping across electrodes.

## 4. Clarification of Claims and Definitions

* **Generalization of Architecture (Section 3.3)**: We have moderated our claims regarding SFM architectures. We now frame the differences between Whisper, HuBERT, and WavLM as hypotheses specific to these models, rather than broad generalizations about their architectures.
* **Terminology**: We have removed causal language (e.g., "mechanistic") and clearly defined the "systematic trade-off" between acoustic precision and higher-level linguistic integration.

We believe these revisions have significantly improved the clarity, robustness, and context of our paper. We address each reviewer's specific comments in detail below.

---

### Decision · Action_Editor_jmAD · 2026-03-28

**Recommendation:** Reject

**Audience:**

Yes

**Audience Explanation:**

The empirical findings of this work may be of interest to researchers working on speech and ECoG data. However, the methodological novelty is limited, as the adopted approaches have largely been explored in prior work.

**Claims And Evidence:**

No

**Claims Explanation:**

The reviewers reached a consensus that the submission contains several critical issues that have not been fully addressed in the authors’ responses.

1. One reviewer noted that the analyses and variance-partitioning methods appear to largely replicate approaches that used in prior studies, rather than offering a clear methodological or conceptual advancement.

2. Several claims—particularly those related to interpretability—are considered to be overstated and not adequately supported by the presented analyses. The use of “interpretable features,” including low-level and linguistic feature regressions, is standard practice in the field and is typically employed as control variables to isolate higher-level representations, rather than as primary evidence of interpretability.

3. One reviewer emphasized that several suggested analyses (e.g., noise ceilings, robustness checks, and feature correlation analyses) are not merely optional extensions for future work, but are essential components needed to substantiate the claims and meet the bar for acceptance.

Based on these assessments, the Action Editor concludes that the submission is not yet ready for publication.

**Resubmission Of Major Revision:**

The authors may consider submitting a major revision at a later time.